# SpiderBoost and Momentum: Faster Stochastic Variance Reduction Algorithms

**Zhe Wang**
Department of ECE
The Ohio State University
wang.10982@osu.edu

**Kaiyi Ji**
Department of ECE
The Ohio State University
ji.367@osu.edu

**Yi Zhou**
Department of ECE
The University of Utah
yi.zhou@utah.edu

**Yingbin Liang**
Department of ECE
The Ohio State University
liang.889@osu.edu

**Vahid Tarokh**
Department of ECE
Duke University
vahid.tarokh@duke.edu

## Abstract

SARAH and SPIDER are two recently developed stochastic variance-reduced algorithms, and SPIDER has been shown to achieve a near-optimal first-order oracle complexity in smooth nonconvex optimization. However, SPIDER uses an accuracy-dependent stepsize that slows down the convergence in practice, and cannot handle objective functions that involve nonsmooth regularizers. In this paper, we propose SpiderBoost as an improved scheme, which allows to use a much larger constant-level stepsize while maintaining the same near-optimal oracle complexity, and can be extended with proximal mapping to handle composite optimization (which is nonsmooth and nonconvex) with provable convergence guarantee. In particular, we show that proximal SpiderBoost achieves an oracle complexity of $\mathcal{O}(\min\{n^{1/2}\epsilon^{-2}, \epsilon^{-3}\})$ in composite nonconvex optimization, improving the state-of-the-art result by a factor of $\mathcal{O}(\min\{n^{1/6}, \epsilon^{-1/3}\})$. We further develop a novel momentum scheme to accelerate SpiderBoost for composite optimization, which achieves the near-optimal oracle complexity in theory and substantial improvement in experiments.

## 1 Introduction

We consider the following finite-sum optimization problem

$$\min_{x \in \mathbb{R}^d} \Psi(x) := f(x), \text{ where } f(x) := \frac{1}{n} \sum_{i=1}^{n} f_i(x) \tag{P}$$

where the function $f$ denotes the total loss on the training samples and in general is nonconvex. Since large-scale machine learning problems can have very large sample size $n$, the full-batch gradient descent algorithm has high computational complexity. Thus, various stochastic gradient descent (SGD) algorithms have been proposed. For nonconvex optimization, the basic SGD algorithm, which calculates the gradient of one data sample per iteration, has been shown to yield an overall stochastic first-order oracle (SFO) complexity, i.e., gradient complexity, of $\mathcal{O}(\epsilon^{-4})$ [9] to attain a first-order stationary point $\bar{x}$ that satisfies $\mathbb{E}\|\nabla f(\bar{x})\| \leq \epsilon$. SGD algorithms with diminishing step-size [9, 5] or a sufficiently large batch size [37, 11] were also proposed to guarantee their convergence to a stationary point rather than its neighborhood.

Furthermore, various variance reduction methods have been proposed, which construct more accurate stochastic gradient estimators than that of SGD, e.g., SAG [31], SAGA [7] and SVRG [16]. In

particular, SAGA and SVRG have been shown to yield an overall SFO complexity of $O(n^{2/3}\epsilon^{-2})$ [29, 4]. Recently, [24, 25] proposed a variance reduction method called SARAH, where the gradient estimator is sequentially updated in the inner loop to improve the estimation accuracy. In particular, SARAH has been shown in [25] to achieve an overall $\mathcal{O}(\epsilon^{-4})$ SFO complexity for nonconvex optimization. Another variance reduction method called SPIDER was proposed in [8], which uses the same gradient estimator as that of SARAH but adopts a normalized gradient update with a stepsize $\eta = \mathcal{O}(\epsilon/L)$. [8] showed that SPIDER achieves an overall $\mathcal{O}(\min\{n^{1/2}\epsilon^{-2}, \epsilon^{-3}\})$ SFO, which was further shown to be optimal in the regime with $n \leq \mathcal{O}(\epsilon^{-4})$.

Though SPIDER is theoretically appealing, three important issues still require further attention. First, SPIDER requires a very restrictive stepsize $\eta = \mathcal{O}(\epsilon/L)$ to guarantee its convergence, which prevents SPIDER from making big progress even if it is possible. Relaxing such a condition appears not easy under its original convergence analysis framework.

- *This paper proposes a more practical SpiderBoost algorithm, which allows a much larger stepsize $\eta = \mathcal{O}(1/L)$ than SPIDER while retaining the same state-of-the-art complexity order as SPIDER (see Table 2 in Suppl). This is due to the new convergence analysis idea that we develop, which analyzes the increments of variables over each entire inner loop rather than over each inner-loop iteration, and hence yields tighter bound and consequently more relaxed stepsize requirement.*

Second, the convergence analysis of SPIDER requires a very small per-iteration increment $\|x_{k+1} - x_k\| = \mathcal{O}(\epsilon/L)$, which is difficult to guarantee if one attempts to generalize it to a proximal algorithm for solving the composite optimization problem (see Section 3) that possibly involves nonsmoothness. Hence, generalizing SPIDER to the proximal setting with provable convergence guarantee is challenging.

- *Our SpiderBoost has a natural generalization, i.e., the Prox-SpiderBoost algorithm, which can be applied to solve composite optimization problems. We show that Prox-SpiderBoost achieves a SFO complexity of $\mathcal{O}(n^{1/2}\epsilon^{-2})$ and a proximal oracle (PO) complexity of $\mathcal{O}(\epsilon^{-2})$, which improves the existing best results by a factor of $\mathcal{O}(n^{1/6})$ (see Table 1).*

Third, although SPIDER achieves the near-optimal oracle complexity in nonconvex optimization, its practical performance has been found [25, 8] to be hardly advantageous over SVRG. Therefore, it is of vital importance to exploit other algorithmic dimensions to further improve the practical performance of SPIDER, and momentum is such a promising perspective. However, the existing analysis of variance-reduced algorithms has been explored for SVRG only in certain *convex* scenarios [27, 1, 2, 32] and under a local gradient dominance geometry in nonconvex optimization [19]. Therefore, it is not even clear whether a certain momentum scheme can be applied to SPIDER and yield the optimal oracle gradient complexity for general nonconvex optimization.

- *This paper proposes a momentum scheme to accelerate the Prox-SpiderBoost, named Prox-SpiderBoost-M, for composite optimization. We show that Prox-SpiderBoost-M achieves an oracle complexity order of $O(n + \sqrt{n}\epsilon^{-2})$, matching the complexity lower bound for nonconvex optimization. In contrast to the existing analysis for stochastic algorithms with momentum [10] for nonconvex optimization, our proof exploits the martingale structure of the gradient estimator to bound the variance term and its accumulations over the entire optimization path in a tight way under the momentum scheme.*

Due to space limitation, we relegate several other results to the supplementary materials, including analysis of Prox-SpiderBoost under non-Euclidean geometry and Polyak-Łojasiewicz condition, and analysis of both Prox-SpiderBoost and Prox-SpiderBoost-M for online nonconvex composite optimization.

## 1.1 Related Work

**Stochastic algorithms for smooth nonconvex optimization:** The convergence analysis for SGD was studied in [11] for smooth nonconvex optimization. SGD with diminishing stepsize and sufficiently large batch size were further studied in [11, 5, 37] to improve the performance. Various variance-reduced algorithms have been proposed and studied, including, e.g., SAG [31], SAGA [7], SVRG [16, 29, 4], SCSG [18], SNVRG [36], SARAH [24, 25, 26, 28], SPIDER [8]. In particular, SPIDER has been shown in [8] to achieve the oracle complexity lower bound for a certain regime. Such an idea has also been extended for optimization over manifolds in [38, 35], zeroth-order optimization in [15], ADMM in [12], zeroth-order ADMM in [13], problem with nonsmooth nonconvex

Table 1: Comparison of SFO complexity and PO complexity for composite optimization.

| Algorithms | | Stepsize $\eta$ | Finite-Sum | | Finite-Sum/Online[1] | |
|---|---|---|---|---|---|---|
| | | | SFO | PO | SFO | PO |
| ProxGD | [11] | $\mathcal{O}(L^{-1})$ | $\mathcal{O}(n\epsilon^{-2})$ | $\mathcal{O}(\epsilon^{-2})$ | N/A | N/A |
| ProxSGD | [11] | $\mathcal{O}(L^{-1})$ | N/A | N/A | $\mathcal{O}(\epsilon^{-4})$ | $\mathcal{O}(\epsilon^{-2})$ |
| ProxSVRG/SAGA | [30] | $\mathcal{O}(L^{-1})$ | $\mathcal{O}(n+n^{2/3}\epsilon^{-2})$ | $\mathcal{O}(\epsilon^{-2})$ | N/A | N/A |
| Natasha1.5 | [3] | $\mathcal{O}(\epsilon^{2/3}L^{-2/3})$ | N/A | N/A | $\mathcal{O}(\epsilon^{-3}+\epsilon^{-10/3})$ | $\mathcal{O}(\epsilon^{-10/3})$ |
| ProxSVRG+ | [22] | $\mathcal{O}(L^{-1})$ | $\mathcal{O}(n+n^{2/3}\epsilon^{-2})$ | $\mathcal{O}(\epsilon^{-2})$ | $\mathcal{O}(\epsilon^{-10/3})$ | $\mathcal{O}(\epsilon^{-2})$ |
| Prox-SpiderBoost | (This Work) | $\mathcal{O}(L^{-1})$ | $\mathcal{O}(n+n^{1/2}\epsilon^{-2})$ | $\mathcal{O}(\epsilon^{-2})$ | $\mathcal{O}(\epsilon^{-2}+\epsilon^{-3})$ | $\mathcal{O}(\epsilon^{-2})$ |

[1] The online setting refers to the case where the objective function takes the form of the expected value of the loss function over the data distribution. Such a method can also be applied to solve the finite-sum problem, and hence the SFO complexity in the last column is applicable to both the finite-sum and online problems. Thus, for algorithms that have SFO bounds available in both of the last two columns, the minimum between the two bounds provides the best bound for the finite-sum problem.

regularizer in [33], stochastic composite optimization in [34], noisy gradient descent in [21], and an adaptive batch size scheme in [14]. Our study here proposes a SpiderBoost algorithm, which substantially improves the stepsize of SPIDER while retaining the same performance guarantee and performs much faster than SPIDER in practice.

**Stochastic algorithms for composite nonconvex optimization:** Proximal SGD has been proposed and studied by [9, 10] to solve composite nonconvex optimization problems. Moreover, variance reduced algorithms such as Prox-SVRG and Prox-SAGA [30], Natasha1.5 [3], and ProxSVRG+ [22] have also been proposed to further improve the performance. Our study proposes Prox-SpiderBoost, which order-level outperforms all the existing algorithms for composite nonconvex optimization.

**Momentum schemes for nonconvex optimization:** For nonconvex optimization, [10] established convergence of SGD with momentum to an $\epsilon$-first-order stationary point with an oracle complexity of $O(\epsilon^{-4})$. The convergence guarantee of SVRG with momentum has been explored under a certain local gradient dominance geometry in nonconvex optimization [19]. Here, we propose Prox-SpiderBoost-M which achieves the complexity lower bound for a certain regime, and practically substantially outperforms existing variance reduced algorithms with momentum.

## 2 SpiderBoost for Nonconvex Optimization

### 2.1 SpiderBoost Algorithm

In this section, we introduce the SpiderBoost algorithm designed for the problem (P). In [24], a novel gradient estimator was introduced for reducing the variance. More specifically, consider a certain inner loop $\{x_k\}_{k=0}^{q-1}$. The initialization of the estimator is set to be $v_0 = \nabla f(x_0)$. Then, for each subsequent iteration $k$, an index set $S$ is sampled and the corresponding estimator $v_k$ is constructed as

$$v_k = \frac{1}{|S|}\sum_{i\in S}\big[\nabla f_i(x_k) - \nabla f_i(x_{k-1}) + v_{k-1}\big]. \tag{1}$$

It can be seen that the estimator in eq. (1) is constructed iteratively based on the information $x_{k-1}$ and $v_{k-1}$ that are obtained from the previous update. As a comparison, the SVRG estimator [16] is constructed based on the information of the initialization of that loop (i.e., replace $x_{k-1}$ and $v_{k-1}$ in eq. (1) with $x_0$ and $v_0$, respectively). Therefore, the estimator in eq. (1) utilizes more fresh information and yields more accurate estimation of the full gradient. The estimator in eq. (1) has been adopted by [24, 25] and [8] for proposing SARAH and SPIDER, respectively. In specific, SPIDER was shown in [8] to be optimal in the regime with $n \leq \mathcal{O}(\epsilon^{-4})$.

Though SPIDER has desired performance in theory, it can run very slowly in practice due to the choice of a conservative stepsize. In specific, SPIDER uses a very small stepsize $\eta = \mathcal{O}(\frac{\epsilon}{L})$ (where $\epsilon$ is the desired accuracy) in normalized gradient descent, which yields small increment per iteration, i.e., $\|x_{k+1} - x_k\| = \mathcal{O}(\epsilon)$. By following the analysis of SPIDER, such a stepsize appears to be necessary in order to achieve the desired convergence rate.

| **Algorithm 1** SpiderBoost | **Algorithm 2** Prox-SpiderBoost |
|---|---|
| **Input:** $\eta = \frac{1}{2L}, q, K, |S| \in \mathbb{N}$. | **Input:** $\eta = \frac{1}{2L}, q, K, |S| \in \mathbb{N}$. |
| **for** $k = 0, 1, \ldots, K - 1$ **do** | **for** $k = 0, 1, \ldots, K - 1$ **do** |
| $\quad$ **if** $mod(k, q) = 0$ **then** | $\quad$ **if** $mod(k, q) = 0$ **then** |
| $\quad\quad$ Compute $v_k = \nabla f(x_k)$, | $\quad\quad$ Compute $v_k = \nabla f(x_k)$, |
| $\quad$ **else** | $\quad$ **else** |
| $\quad\quad$ Draw $|S|$ samples with replacement. | $\quad\quad$ Draw $|S|$ samples with replacement. |
| $\quad\quad$ Compute $v_k$ according to eq. (1). | $\quad\quad$ Compute $v_k$ according to eq. (1). |
| $\quad$ **end** | $\quad$ **end** |
| $\quad x_{k+1} = x_k - \eta v_k$. | $\quad x_{k+1} = \mathrm{prox}_{\eta h}(x_k - \eta v_k)$. |
| **end** | **end** |
| **Output:** $x_\xi$, where $\xi \overset{\mathrm{Unif}}{\sim} \{0, \ldots, K - 1\}$. | **Output:** $x_\xi$, where $\xi \overset{\mathrm{Unif}}{\sim} \{0, \ldots, K - 1\}$. |

Such a conservative stepsize adopted by SPIDER motivates our design of an improved algorithm named SpiderBoost (see Algorithm 1), which uses the same estimator eq. (1) as SARAH and SPIDER, but adopts a much larger stepsize $\eta = \frac{1}{2L}$, as opposed to $\eta = \mathcal{O}(\frac{\epsilon}{L})$ taken by SPIDER. Also, SpiderBoost updates the variable via a gradient descent step (same as SARAH), as opposed to the normalized gradient descent step taken by SPIDER. Furthermore, SpiderBoost generates the output variable via a random strategy whereas SPIDER outputs deterministically. Collectively, SpiderBoost can make a considerably larger progress per iteration than SPIDER, especially in the initial optimization phase where the estimated gradient norm $\|v_k\|$ is large, and is still guaranteed to achieve the same desirable convergence rate as SPIDER, as we show in the next subsection. We compare the empirical performance between SPIDER and SpiderBoost in Section 5.1.

## 2.2 Convergence Analysis of SpiderBoost

In this subsection, we study the convergence rate and complexity of SpiderBoost. In particular, we adopt the following standard assumptions.

**Assumption 1.** *The objective function in the problem (P) satisfies:*
*1. The object function $\Psi$ is bounded below, i.e., $\Psi^* := \inf_{x \in \mathbb{R}^d} \Psi(x) > -\infty$;*

*2. Each gradient $\nabla f_i, i = 1, ..., n$ is $L$-Lipschitz continuous, i.e., $\forall x, y \in \mathbb{R}^d, \|\nabla f_i(x) - \nabla f_i(y)\| \leq L\|x - y\|$.*

Assumption 1 essentially assumes that the smooth objective function has a non-trivial minimum and its gradient is Lipschitz continuous, which are valid and standard conditions in nonconvex optimization. Then, we obtain the following convergence result for SpiderBoost.

**Theorem 1.** *Let Assumption 1 hold and apply SpiderBoost in Algorithm 1 to solve the problem (P) with parameters $q = |S| = \sqrt{n}$ and stepsize $\eta = \frac{1}{2L}$. Then, the corresponding output $x_\xi$ satisfies $\mathbb{E}\|\nabla f(x_\xi)\| \leq \epsilon$ provided that the total number $K$ of iterations satisfies*

$$K \geq \mathcal{O}\Big(\frac{L(f(x_0) - f^*)}{\epsilon^2}\Big).$$

*Moreover, the overall SFO complexity is $\mathcal{O}(\sqrt{n}\epsilon^{-2} + n)$.*

Theorem 1 shows that the output of SpiderBoost achieves the first-order stationary condition within $\epsilon$ accuracy with a total SFO complexity $\mathcal{O}(\sqrt{n}\epsilon^{-2} + n)$. This matches the lower bound that one can expect for first-order algorithms in the regime $n \leq \mathcal{O}(\epsilon^{-4})$ [8]. As we explain in Section 2.1, SpiderBoost enhances SPIDER mainly due to the utilization of a large constant stepsize, which yields significant acceleration over SPIDER in practice as we illustrate in the experiments in Section 5.1.

We note that the analysis of SpiderBoost in Theorem 1 is very different from that of SPIDER that depends on an $\epsilon$-level stepsize and the normalized gradient descent step to guarantee a constant increment $\|x_{k+1} - x_k\|$ in every iteration. In contrast, SpiderBoost exploits the special structure of gradient estimator and analyzes the algorithm over the entire inner loop rather than over each iteration, and thus yields a better bound.

# 3 Prox-SpiderBoost for Nonconvex Composite Optimization

In this section, we generalize SpiderBoost to solve the following nonconvex composite problem:

$$\min_{x \in \mathcal{X}} \Psi(x) := f(x) + h(x), \quad f(x) := \frac{1}{n} \sum_{i=1}^{n} f_i(x) \tag{Q}$$

where the function $f$ is possibly nonconvex, $h$ is a simple convex but possibly nonsmooth regularizer, and $\mathcal{X}$ is a convex constrained set. To handle the nonsmoothness, we next introduce the proximal mapping which is an effective tool for composite optimization.

## 3.1 Preliminaries on Proximal Mapping

Consider a proper and lower-semicontinuous function $h$ (which can be non-differentiable). We define its proximal mapping at $x \in \mathbb{R}^d$ with parameter $\eta > 0$ as

$$\text{prox}_{\eta h}(x) := \arg \min_{u \in \mathcal{X}} \left\{ h(u) + \frac{1}{2\eta} \|u - x\|^2 \right\}.$$

Such a mapping is well defined and is unique particularly for convex functions. Furthermore, the proximal mapping can be used to generalize the first-order stationary condition of smooth optimization to nonsmooth composite optimization via the following fact.

**Fact 1.** *Let $h$ be a proper and convex function. Define the following notion of generalized gradient*

$$G_\eta(x) := \frac{1}{\eta} \Big( x - \text{prox}_{\eta h}(x - \eta \nabla f(x)) \Big). \tag{2}$$

*Then, $x$ is a critical point of $\Psi := f + h$ (i.e., $0 \in \nabla f(x) + \partial h(x)$) if and only if $G_\eta(x) = 0$.*

Fact 1 introduces a generalized notion of gradient for composite optimization. To elaborate, consider the case $h \equiv 0$ so that the proximal mapping becomes the identity mapping. Then, the generalized gradient $G_\eta(x)$ reduces to the gradient $\nabla f(x)$ of the unconstrained optimization. Therefore, the $\epsilon$-first-order stationary condition for composite optimization is naturally defined as $\|G_\eta(x)\| \leq \epsilon$.

## 3.2 Prox-SpiderBoost and Oracle Complexity

To generalize to composite optimization, SpiderBoost admits a natural extension Prox-SpiderBoost, whereas SPIDER encounters challenges. The main reason is because SpiderBoost admits a constant stepsize and its convergence guarantee does not have any restriction on the per-iteration increment of the variable. However, the convergence of SPIDER requires the per-iteration increment of the variable to be at the $\epsilon$-level, which is challenging to satisfy under the nonlinear proximal operator in composite optimization.

The detailed steps of Prox-SpiderBoost (which generalizes SpiderBoost to composite optimization objectives) are described in Algorithm 2. In particular, Prox-SpiderBoost updates the variable via a proximal gradient step to handle the possible nonsmoothness in composite optimization.

We next characterize the oracle complexity of Prox-SpiderBoost for achieving the generalized $\epsilon$-first-order stationary condition.

**Theorem 2.** *Let Assumption 1 hold and consider the problem (Q) with $\mathcal{X} = \mathbb{R}^d$. Apply the Prox-SpiderBoost in Algorithm 2 with parameters $q = |S| = \sqrt{n}$ and $\eta = \frac{1}{2L}$. Then, the corresponding output $x_\xi$ satisfies $\mathbb{E}\|G_\eta(x_\xi)\| \leq \epsilon$ provided that the total number $K$ of iterations satisfies*

$$K \geq \mathcal{O}\Big( \frac{L(\Psi(x_0) - \Psi^*)}{\epsilon^2} \Big).$$

*Moreover, the SFO complexity is $\mathcal{O}(\sqrt{n}\epsilon^{-2} + n)$, and the proximal oracle (PO) complexity is $\mathcal{O}(\epsilon^{-2})$.*

As a comparison, the SFO complexity $\mathcal{O}(\sqrt{n}\epsilon^{-2} + n)$ of Prox-SpiderBoost in Theorem 2 improves the existing complexity result by a factor of $n^{1/6}$ [22]. Furthermore, the complexity lower bound for achieving the $\epsilon$-first-order stationary condition in un-regularized optimization [8] also serves as a lower bound for composite optimization (by considering the special case $h \equiv 0$). Therefore, the

SFO complexity of our Prox-SpiderBoost matches the corresponding complexity lower bound in the regime with $n \leq \mathcal{O}(\epsilon^{-4})$, and is hence near optimal.

Moreover, our Prox-SpiderBoost still achieves the state-of-the-art convergence results under other settings such as online optimization, non-Euclidean geometry and Polyak-Łojasiewicz condition. Due to the space limitation, we relegate these results to Appendix D.

# 4 Accelerating Prox-SpiderBoost via Momentum

In this section, we propose a proximal SpiderBoost algorithm that incorporates a momentum scheme (referred to as Prox-SpiderBoost-M) for solving the composite problem (Q), and study its theoretical guarantee as well as the oracle complexity.

## 4.1 Algorithm Design

We present the detailed update rule of Prox-SpiderBoost-M in Algorithm 3.

---

**Algorithm 3** Prox-SpiderBoost-M

**Input:** $q, K \in \mathbb{N}, \{\lambda_k\}_{k=1}^{K-1}, \{\beta_k\}_{k=1}^{K-1} > 0, y_0 = x_0 \in \mathbb{R}^d$, and set $\alpha_k = \frac{2}{\lceil k/q \rceil + 1}$.

**for** $k = 0, 1, \ldots, K-1$ **do**

    $z_k = (1 - \alpha_{k+1})y_k + \alpha_{k+1}x_k$,

    **if** $mod(k, q) = 0$ **then**

        set $v_k = \nabla f(z_k)$,

    **else**

        Draw $\xi_k$ samples with replacement and compute $v_k$ according to eq. (1).

    **end**

    $x_{k+1} = \text{prox}_{\lambda_k h}(x_k - \lambda_k v_k)$,

    $y_{k+1} = z_k - \frac{\beta_k}{\lambda_k} x_k + \frac{\beta_k}{\lambda_k} \text{prox}_{\lambda_k h}(x_k - \lambda_k v_k)$.

**end**

**Output:** $z_\zeta$, where $\zeta \overset{\text{Unif}}{\sim} \{0, \ldots, K-1\}$.

---

To elaborate on the algorithm design, note that Prox-SpiderBoost-M generates a tuple of variable sequences $\{x_k, y_k, z_k\}_k$ according to the momentum scheme. In specific, the variables $x_k, y_k$ are updated via proximal gradient-like steps using the gradient estimate $v_k$ proposed for SARAH in [24, 25] and different stepsizes $\lambda_k, \beta_k$, respectively. Then, their convex combination with momentum coefficient $\alpha_{k+1}$ yields the variable $z_{k+1}$. Here, we choose a standard momentum coefficient scheduling that diminishes epochwisely (see the expression for $\alpha_k$) for proving convergence guarantee in nonconvex optimization. We also note that the two updates for $x_{k+1}$ and $y_{k+1}$ do not introduce extra computation overhead as compared to a single update, since they both depend on the same proximal term.

We want to highlight the difference between our momentum scheme for Prox-SpiderBoost-M and the existing momentum scheme design for proximal SGD in [10] and proximal SVRG in [1]. In these works, they use the following proximal gradient steps for updating the variables $x_{k+1}$ and $y_{k+1}$:

$$x_{k+1} = \text{prox}_{\lambda_k h}(x_k - \lambda_k v_k), \qquad y_{k+1} = \text{prox}_{\beta_k h}(z_k - \beta_k v_k). \tag{3}$$

Note that eq. (3) use different proximal updates that are based on $x_k$ and $z_k$, respectively. As a comparison, our momentum scheme in Algorithm 3 applies the same proximal gradient term $\text{prox}_{\lambda_k h}(x_k - \lambda_k v_k)$ to update both variables $x_{k+1}$ and $y_{k+1}$, and therefore requires less computation. Moreover, our update for the variable $y_{k+1}$ is not a single proximal gradient update (as opposed to eq. (3)), and it couples with the variables $z_k$ and $x_k$.

The momentum scheme introduced in [1] was not proven to have a convergence guarantee in nonconvex optimization. In the next subsection, we prove that our momentum scheme in Algorithm 3 has a provable convergence guarantee for nonconvex composite optimization with convex regularizers.

## 4.2 Convergence and Complexity Analysis

In this subsection, we study the convergence guarantee of Prox-SpiderBoost-M for solving the problem (Q). We obtain the following main result.

**Theorem 3.** *Let Assumption 1 hold. Apply Prox-SpiderBoost-M (see Algorithm 3) to solve the problem (Q) with parameters $q = |\xi_k| \equiv \sqrt{n}$, $\beta_k \equiv \frac{1}{8L}$ and $\lambda_k \in [\beta_k, (1 + \alpha_k)\beta_k]$. Then, the output $z_\zeta$ produced by the algorithm satisfies $\mathbb{E}\|G_{\lambda_\zeta}(z_\zeta, \nabla f(z_\zeta))\| \leq \epsilon$ for any $\epsilon > 0$ provided that the total number $K$ of iterations satisfies*

$$K \geq \mathcal{O}\left(\frac{L(\Psi(x_0) - \Psi^*)}{\epsilon^2}\right). \tag{4}$$

*Moreover, the SFO complexity is at most $\mathcal{O}(n + \sqrt{n}\epsilon^{-2})$ and the PO complexity is at most $\mathcal{O}(\epsilon^{-2})$.*

Theorem 3 establishes the convergence rate of Prox-SpiderBoost-M to satisfy the generalized first-order stationary condition and the corresponding oracle complexity. Specifically, the iteration complexity to achieve the generalized $\epsilon$-first-order stationary condition is in the order of $\mathcal{O}(\epsilon^{-2})$, which matches that of Prox-SpiderBoost. Furthermore, the corresponding SFO complexity $\mathcal{O}(n + \sqrt{n}\epsilon^{-2})$ matches the lower bound for nonconvex optimization [8]. Therefore, Prox-SpiderBoost-M enjoys the same optimal convergence guarantee as that for the Prox-SpiderBoost in nonconvex optimization, and it further benefits from the momentum scheme that can lead to significant acceleration in practical applications (as we demonstrate via experiments in Section 5).

From a technical perspective, we highlight the following three major new developments in the proof of Theorem 3 that is different from the proof for the basic stochastic gradient algorithm with momentum [10] for nonconvex optimization: 1) our proof exploits the martingale structure of the SPIDER estimate $v_k$ which allows to bound the mean-square error term $\mathbb{E}\|\nabla f(z_k) - v_k\|^2$ in a tight way under the momentum scheme. In traditional analysis of stochastic algorithms with momentum [10], such an error term corresponds to the variance of the stochastic estimator and is assumed to be bounded by a universal constant. 2): Our proof requires a very careful manipulation of the bounding strategy to handle the accumulation of the mean-square error $\mathbb{E}\|\nabla f(z_k) - v_k\|^2$ over the entire optimization path.

## 5 Experiments

### 5.1 Comparison between SpiderBoost and SPIDER

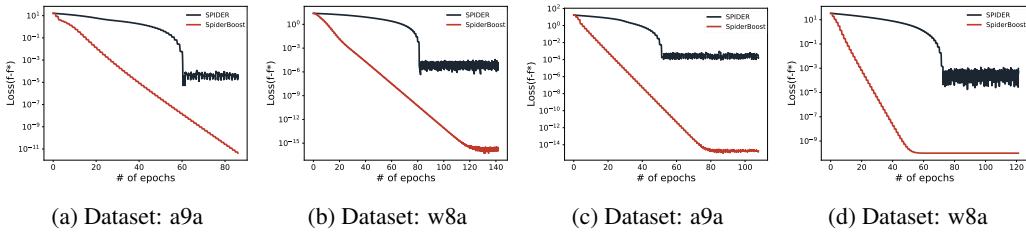

(a) Dataset: a9a     (b) Dataset: w8a     (c) Dataset: a9a     (d) Dataset: w8a

Figure 1: (a) and (b): Logistic regression problem with nonconvex regularizer. (c) and (d): Robust linear regression problem with an $l_2$ regularizer.

In this subsection, we compare the performance of SPIDER and SpiderBoost for solving the logistic regression problem with a nonconvex regularizer and the nonconvex robust linear regression problem (See Appendix F for the forms of the objective functions). For each problem, we apply two different datasets from the LIBSVM [6]: the a9a dataset ($n = 32561, d = 123$) and the w8a dataset ($n = 49749, d = 300$). For both algorithms, we use the same parameter setting except for the stepsize. As specified in [8] for SPIDER, we set $\eta = 0.01$ (determined by a prescribed accuracy to guarantee convergence). On the other hand, SpiderBoost allows to set $\eta = 0.05$. Figure 1 shows the convergence of the function value gap of both algorithms versus the number of passes that are taken over the data. It can be seen that SpiderBoost enjoys a much faster convergence than that of SPIDER due to the allowance of a large stepsize. Furthermore, SPIDER oscillates around a point, which is the prescribed accuracy that determines the adopted stepsize $\eta = 0.01$. This implies that setting a larger stepsize for SPIDER would cause it to saturate and start to oscillate at a certain function value, which is undesired.

### 5.2 Comparison of SpiderBoost Type of Algorithms with Other Algorithms

In this subsection, we compare the performance of our SpiderBoost (for smooth problems), Prox-SpiderBoost (for composite problems), and Prox-SpiderBoost-M with other existing stochastic

variance-reduced algorithms including SVRG in [16], Katyusha$^{ns}$ in [1], ASVRG in [32], RSAG in [10]. We note that all algorithms use certain momentum schemes except for SVRG, SpiderBoost, and Prox-SpiderBoost. For all algorithms considered, we set their learning rates to be $0.05$. For each experiment, we initialize all the algorithms at the same point that is generated randomly from the normal distribution. Also, we choose a fixed mini-batch size $256$ and set the epoch length $q$ to be $2n/256$ such that all algorithms pass over the entire dataset twice in each epoch.

We first apply these algorithms to solve two smooth nonconvex problems: logistic regression and robust linear regression problems, each with datasets of a9a and w8a, and report the experiment results in Figure 2. One can see from Figure 2 that our Prox-SpiderBoost-M achieves the best performance and significantly outperforms other algorithms. Also, the performances of both Katyusha$^{ns}$ and ASVRG do not achieve much acceleration in such a nonconvex case, as these algorithms are originally developed to achieve acceleration for convex problems. This demonstrates that our design of Prox-SpiderBoost-M has a stable performance in nonconvex optimization as well as provable theoretical guarantee. We note that the curve of SpiderBoost overlaps with that of SVRG similarly to the results reported in other recent studies.

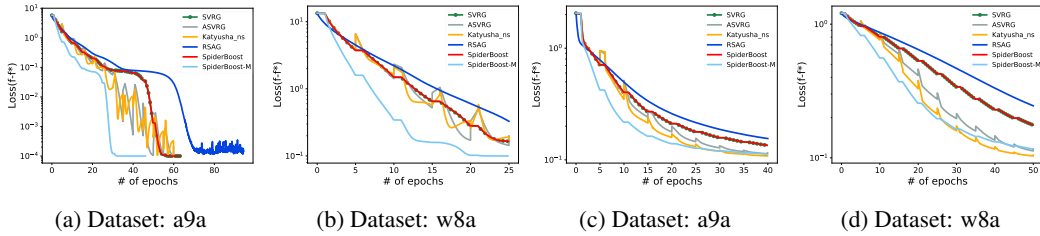

| (a) Dataset: a9a | (b) Dataset: w8a | (c) Dataset: a9a | (d) Dataset: w8a |

Figure 2: (a) and (b): Logistic regression with nonconvex regularizer, (c) and (d): Robust linear regression..

We further add an $\ell_1$ nonsmooth regularizer with weight coefficient $0.1$ to the objective functions of the above two optimization problems, and apply the corresponding proximal versions of these algorithms to solve the nonconvex composite optimization problems. All the results are presented in Figures 3. One can see that our Prox-SpiderBoost-M still significantly outperforms all the other algorithms in these nonsmooth and nonconvex scenarios. This demonstrates that our novel design of the coupled update for $\{y_k\}_k$ in the momentum scheme is efficient in the nonsmooth and nonconvex setting. Also, it turns out that Katyusha$^{ns}$ and ASVRG are suffering from a slow convergence (their convergences occur at around 40 epochs). Together with the above experiments for smooth problems, this implies that their performance is not stable and may not be generally suitable for solving nonconvex problems.

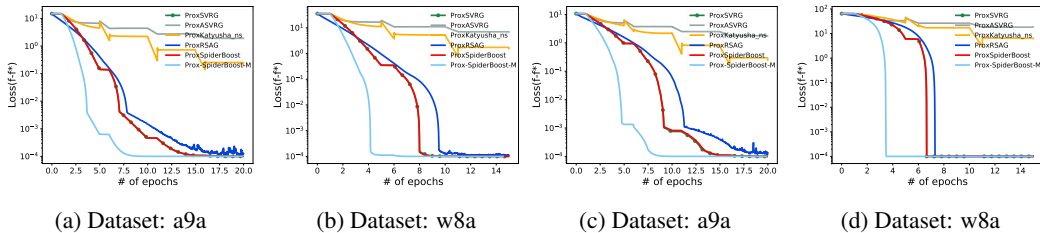

| (a) Dataset: a9a | (b) Dataset: w8a | (c) Dataset: a9a | (d) Dataset: w8a |

Figure 3: (a) and (b): Logistic regression with an $\ell_1$ nonsmooth regualarizer. (c) and (d): Robust linear regression with an $\ell_1$ nonsmooth regualarizer.

## 6 Conclusion

In this paper, we proposed the SpiderBoost algorithm, which achieves the same near-optimal complexity performance as SPIDER, but allows a much larger stepsize and hence runs faster in practice than SPIDER. We then extend the proposed SpiderBoost to solve composite nonconvex optimization, and proposed a momentum scheme to further accelerate the algorithm. For all these algorithms, we develop new techniques to characterize the performance bounds, all of which achieve the best state-of-the-art. We anticipate that SpiderBoost has a great potential to be applied to various other large-scale optimization problems.

## Acknowledgments

The work of Z. Wang, K. Ji, and Y. Liang was supported in part by the U.S. National Science Foundation under the grants CCF-1761506, CCF-1909291, and CCF-1900145.

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
