[Supplementary Material]

# Supplementary Materials

## A  Comparison of SFO Complexity for Smooth Nonconvex Optimization

Table 2: Comparison of SFO complexity for smooth nonconvex optimization

| Algorithms | | Stepsize $\eta$ | Finite-sum SFO | Finite-sum/Online SFO |
|---|---|---|---|---|
| GD | [23] | $\mathcal{O}(L^{-1})$ | $\mathcal{O}(n\epsilon^{-2})$ | N/A[2] |
| SGD | [11] | $\mathcal{O}(L^{-1})$ | N/A | $\mathcal{O}(\epsilon^{-4})$ |
| SVRG | [29] [4] | $\mathcal{O}(L^{-1}n^{-2/3})$ | $\mathcal{O}(n + n^{2/3}\epsilon^{-2})$ | N/A |
| SCSG | [18] | $\mathcal{O}(L^{-1}(n^{-2/3} \wedge \epsilon^{4/3}))$ | $\mathcal{O}(n + n^{2/3}\epsilon^{-2})$ | $\mathcal{O}(\epsilon^{-2} + \epsilon^{-10/3})$ |
| SARAH | [25, 24] | $\mathcal{O}((L\sqrt{q})^{-1})$[3] | N/A | $\mathcal{O}(\epsilon^{-4})$ |
| SNVRG | [36] | $\mathcal{O}(L^{-1})$ | $\mathcal{O}((n + n^{1/2}\epsilon^{-2})\log(n))$ | $\mathcal{O}((\epsilon^{-2} + \epsilon^{-3})\log(\epsilon^{-1}))$ |
| SPIDER | [8] | $\mathcal{O}(\epsilon L^{-1})$ [4] | $\mathcal{O}(n + n^{1/2}\epsilon^{-2})$ | $\mathcal{O}(\epsilon^{-2} + \epsilon^{-3})$ |
| SpiderBoost | (This Work) | $\mathcal{O}(L^{-1})$ | $\mathcal{O}(n + n^{1/2}\epsilon^{-2})$ | $\mathcal{O}(\epsilon^{-2} + \epsilon^{-3})$ |

[2] For deterministic algorithms, the online setting does not exist.
[3] The stepsize $\eta = \mathcal{O}(1/(L\sqrt{q}))$ is chosen in [24] to guarantee the convergence of SARAH.
[4] SPIDER uses the normalized gradient descent, which can also be viewed as the gradient descent with the stepsize $O(\epsilon L^{-1}/\|v_k\|)$.

## B  Prox-SpiderBoost for Constrained Optimization under Non-Euclidean Geometry

Prox-SpiderBoost proposed in Section 3 adopts the proximal mapping that solves an unconstrained subproblem under the $\ell_2$ Euclidean distance. Such a mapping can be further generalized to solve constrained composite optimization under a non-Euclidean geometry.

To elaborate, consider solving the composite optimization problem (Q) subject to a convex constraint set $\mathcal{X}$. We introduce the following Bregman distance $V$ associated with a kernel function $\omega : \mathcal{X} \to \mathbb{R}$ defined as: for all $x, y \in \mathcal{X}$,

$$V(x, y) = \omega(x) - \omega(y) - \langle \nabla\omega(y), x - y \rangle. \tag{5}$$

Here, the function $\omega$ is smooth and $\alpha$-strongly convex with respect to a certain generic norm. The specific choice of the kernel function $\omega$ should be compatible to the underlying geometry of the constraint set. For example, for the unconstrained case, one can choose $\omega(x) = \frac{1}{2}\|x\|^2$ so that $V(x, y) = \frac{1}{2}\|x - y\|^2$, which is 1-strongly convex with regard to the $\ell_2$-norm, whereas for the simplex constraint set, one can choose $\omega(x) = \sum_{i=1}^{d}(x_i \log x_i - x_i)$ that yields the KL relative entropy distance $V(x, y) = \sum_{i=1}^{d}(x_i \log \frac{x_i}{y_i} + y_i - x_i)$, which is 1-strongly convex with regard to the $\ell_1$-norm.

Based on the Bregman distance, the proximal gradient step in Algorithm 2 can be generalized to the following update rule for solving the constrained composite optimization.

$$T_{\eta h}(x, v) = \arg\min_{u \in \mathcal{X}} \left\{ h(u) + \langle v, u \rangle + \frac{1}{\eta}V(u, x) \right\}. \tag{6}$$

Moreover, the characterization of critical points in Fact 1 remains valid by defining the generalized gradient as $G_\eta(x) = \frac{1}{\eta}(x - T_{\eta h}(x, \nabla f(x)))$. Then, we obtain the following oracle complexity result of Prox-SpiderBoost under the Bregman distance (replace the proximal step in Algorithm 2 by $x_{k+1} = T_{\eta h}(x_k, v_k)$ ) for solving constrained composite optimization.

**Theorem 4.** *Let Assumption 1 hold and consider the problem (Q). Apply Prox-SpiderBoost with a proper Bregman distance $V$ that is $\alpha$-strongly convex, where $\alpha > 7/8$. Choose the parameters $q = |S| = \sqrt{n}$ and $\eta = \frac{1}{2L}$. Then, the algorithm outputs a point $x_\xi$ satisfying $\mathbb{E}\|G_\eta(x_\xi)\| \leq \epsilon$ provided that the total number $K$ of iterations satisfies*

$$K = \frac{4L(\Psi(x_0) - \Psi^*)}{\epsilon^2} \left(\alpha - \frac{7}{8}\right)^{-1} \left(1 + \frac{1}{4\alpha}\right).$$

*Moreover, the total SFO complexity is $\mathcal{O}(\sqrt{n}\epsilon^{-2} + n)$, and the PO complexity is $\mathcal{O}(\epsilon^{-2})$.*

## C   Prox-SpiderBoost under Polyak-Łojasiewicz Condition

Despite the nonconvexity geometry, many machine learning problems have been shown to satisfy the so-called Polyak-Łojasiewicz condition such as phase retrieval [40], blind deconvolution [20] and neural networks [39], etc. This motivates us to explore the theoretical performance of the Prox-SpiderBoost for solving the composite optimization problem (Q) under the generalized Polyak-Łojasiewicz geometry we define below, where the function can still be nonconvex.

**Definition 1.** *Let $x^*$ be a minimizer of function $\Psi = f + h$. Then, $\Psi$ is said to satisfy the Polyak-Łojasiewicz condition with parameter $\tau$ if for all $x \in \mathbb{R}^d$ and $\eta > 0$ one has*

$$\Psi(x) - \Psi(x^*) \leq \tau\|G_\eta(x)\|^2,$$

*where $G_\eta(x)$ is the generalized gradient defined in Fact 1.*

Definition 1 generalizes the traditional Polyak-Łojasiewicz condition for single smooth objective functions to composite objective functions. In particular, such a condition allows the objective function to be nonsmooth and nonconvex, and it requires the growth of the function value to be controlled by the gradient norm.

---

**Algorithm 4** Prox-SpiderBoost-PL

**Input:** $x_0 \in \mathbb{R}^d, q \in \mathbb{N}, \eta < \frac{1}{16L}$.
**For** $k = 0, 1, 2, \ldots K - 1$
  **If mod**$(k, q) = 0$:
    Set $x_k = x_\xi$, where $\xi$ is selected from
    $\{k - q + 1, \ldots, k - 2\}$ uniformly at random.
    Compute $v_k = \nabla f(x_k)$,
  **Else:**
    Draw $|S|$ samples with replacement.
    Compute $v_k$ according to eq. (1).
  $x_{k+1} = \text{prox}_{\eta h}(x_k - \eta v_k)$.
**Output:** $x_\xi$ from $\{x_0, \cdot, x_{K-1}\}$ uniformly at random.

---

In order to solve the composite optimization problems under the generalized Polyak-Łojasiewicz condition, we propose a variant of Prox-SpiderBoost, which we refer to as Prox-SpiderBoost-PL, described in Algorithm 4. We note that Prox-SpiderBoost-PL can also be viewed as a generalization of SARAH [25] to a proximal algorithm with further differences lying in a much larger stepsize than that chosen by SARAH and random sampling with replacement for inner loop iterations, as opposed to sampling without replacement taken by SARAH.

Next, we present the convergence rate characterization of Algorithm 4 for solving composite optimization problems under the generalized Polyak-Łojasiewicz condition.

**Theorem 5.** *Let Assumprion 1 hold and apply Prox-SpiderBoost-PL in Algorithm 4 to solve the problem (Q) with $\mathcal{X} = \mathbb{R}^d$. Assume the objective function satisfies the Polyak-Łojasiewicz condition with parameter $\tau$ and set $q = |S| = \Theta(L\tau), \eta = \frac{1}{8L}$. Then, the generated variable sequence satisfies, for all $t = 1, 2, \ldots$*

$$\mathbb{E}\|G_\eta(x_{tq})\|^2 \leq \frac{64\tau L}{q - 2}\mathbb{E}\|G_\eta(x_{(t-1)q})\|^2.$$

*Consequently, the oracle complexity of Algorithm 4 for finding a point $x_\xi$ that satisfies $\mathbb{E}\|G_\eta(x_\xi)\| \leq \epsilon$ is in the order $\mathcal{O}((n + L^2\tau^2)\log\frac{1}{\epsilon})$.*

Theorem 5 shows that Prox-SpiderBoost-PL in Algorithm 4 converges linearly to a stationary point for solving composite optimization problems under the generalized Polyak-Łojasiewicz condition. We compare the SFO complexity in Theorem 5 with those of previous proposed stochastic proximal algorithms in Table 3. Our result outperforms the state-of-art result in the regime of $\tau < n^{2/3}$, which is desirable for solving large data problem (i.e., $n$ is large). Moreover, we note that both the results of ProxSVRG and ProxSVRG+ requires the condition number to satisfy $L\tau \geqslant \sqrt{n}$ , whereas our result of Prox-SpiderBoost-PL does not require the aforementioned condition, and has the most relaxed dependency on $n, L$ and $\tau$ demonstrating the superior performance of Prox-SpiderBoost-PL for optimizing functions under Polyak-Łojasiewicz geometry.

Table 3: Comparison of results on SFO compelxity and PO compelxity under Polyak-Łojasiewicz condition.

| Algorithms | | Stepsize $\eta$ | Finite-Sum | | Additional |
| | | | SFO | PO | Condition |
| --- | --- | --- | --- | --- | --- |
| ProxGD | [17] | $\mathcal{O}(L^{-1})$ | $\mathcal{O}(n\tau \log(1/\epsilon))$ | $\mathcal{O}(\tau \log(1/\epsilon))$ | - |
| ProxSVRG/SAGA | [30] | $\mathcal{O}(L^{-1})$ | $\mathcal{O}((n + n^{2/3}\tau) \log(1/\epsilon))$ | $\mathcal{O}(\tau \log(1/\epsilon))$ | $L\tau \geqslant \sqrt{n}$ |
| ProxSVRG$^+$ | [22] | $\mathcal{O}(L^{-1})$ | $\mathcal{O}(n^{2/3}\tau \log(1/\epsilon))$ | $\mathcal{O}(\tau \log(1/\epsilon))$ | $L\tau \geqslant \sqrt{n}$ |
| Prox-SpiderBoost-PL | (This Work) | $\mathcal{O}(L^{-1})$ | $\mathcal{O}((n + \tau^2) \log(1/\epsilon))$ | $\mathcal{O}(\tau \log(1/\epsilon))$ | - |

For the case with $h = 0$ (i.e., the problem objective reduces to the smooth function $f$), our algorithm achieves a total SFO complexity of $(n + L^2\tau^2) \log(1/\epsilon)$, which is the same as that achieved by SARAH [25]. However, we note that our algorithm allows to use a constant stepsize at the order of $\mathcal{O}(1/L)$, whereas SARAH used a much smaller stepsize at the order of $\mathcal{O}(1/(L\sqrt{q}))$.

# D   Prox-SpiderBoost-O for Online Nonconvex Composite Optimization

In this section, we study the performance of a variant of Prox-SpiderBoost for solving nonconvex composite optimization problems under the online setting.

## D.1   Unconstrained Optimization under Euclidean Geometry

In this subsection, we study the following composite optimization problem.

$$\min_{x \in \mathcal{X}} \Psi(x) := f(x) + h(x), f(x) = \mathbb{E}_{\zeta}[f_{\zeta}(x)]. \tag{R}$$

Here the objective function $\Psi(x)$ consists of a population risk $\mathbb{E}_{\zeta}[f_{\zeta}(x)]$ over the underlying data distribution, a nonsmoooth but simple convex regularizer $h(x)$, and a convex constrain set $\mathcal{X}$. Such a problem can be viewed to have infinite samples as opposed to finite samples in the finite-sum problem (as in the problem (Q)), and the underlying data distribution is typically unknown a priori. Therefore, one cannot evaluate the full-gradient $\nabla f$ over the underlying data distribution in practice. For such a type of problems, we propose a variant of Prox-SpiderBoost, which applies stochastic sampling to estimate the full gradient for initializing the gradient estimator in each inner loop. We refer to this variant as Prox-SpiderBoost-O, the details of which are summarized in Algorithm 5. It can be seen that Prox-SpiderBoost-O in Algorithm 5 draws $|S_1|$ stochastic samples to estimate the full gradient for initializing the gradient estimator. To analyze its performance, we introduce the following standard assumption on variance.

**Assumption 2.** *The variance of stochastic gradients is bounded, i.e., there exists a constant $\sigma > 0$ such that for all $x \in \mathbb{R}^d$ and all random draws of $\zeta$, it holds that $\mathbb{E}_{\zeta} \|\nabla f_{\zeta}(x) - \nabla f(x)\|^2 \leqslant \sigma^2$.*

Under Assumption 2, the variance of a mini-batch gradient with size $|S_1|$ can be bounded by $\mathcal{O}(\sigma^2/|S_1|)$. We obtain the following result on the oracle complexity for Prox-SpiderBoost-O in Algorithm 5.

**Theorem 6.** *Let Assumptions 1 and 2 hold and consider the problem (R) with $\mathcal{X} = \mathbb{R}^d$. Apply Prox-SpiderBoost-O with parameters $|S_1| = 24\sigma^2\epsilon^{-2}, q = |S| = \sqrt{|S_1|}, \eta = \frac{1}{2L}$. Then, the*

---
**Algorithm 5** Prox-SpiderBoost-O for online optimization
---
**Input:** $\eta = \frac{1}{2L}, q, K, |S_1|, |S| \in \mathbb{N}$.
**For** $k = 0, 1, \ldots, K - 1$
  **If** $\mathrm{mod}(k, q) = 0$:
    Draw $|S_1|$ samples with replacement.
    Set $v_k = \frac{1}{|S_1|} \sum_{i \in S_1} f_i(x_k)$.
  **Else:**
    Draw $|S|$ samples with replacement
    Compute $v_k$ according to eq. (1).
  $x_{k+1} = \mathrm{prox}_{\eta h}(x_k - \eta v_k)$.
**Output:** $x_\xi$ from $\{x_0, \cdot, x_{K-1}\}$ uniformly at random.
---

*corresponding output $x_\xi$ satisfies $\mathbb{E}\|G_\eta(x_\xi)\| \leq \epsilon$ provided that the total number $K$ of iterations satisfies*

$$K \geq \mathcal{O}\Big(\frac{L(\Psi(x_0) - \Psi^*)}{\epsilon^2}\Big).$$

*Moreover, the resulting total SFO complexity is $\mathcal{O}(\epsilon^{-3} + \epsilon^{-2})$, and the PO complexity is $\mathcal{O}(\epsilon^{-2})$.*

To the best of our knowledge, the SFO complexity of Algorithm 5 improves the state-of-art result $\mathcal{O}(\epsilon^{-10/3})$ [22, 3] of online stochastic composite optimization by a factor of $\epsilon^{1/3}$.

In the smooth case with $h(x) = 0$, the problem (R) reduces to the online case of problem (P), and Algorithm 5 reduces to a online version of SpiderBoost. We refer to such an algorithm as SpiderBoost-O. The following corollary characterizes the performance of SpiderBoost-O to solve an online problem.

**Corollary 1.** *Let Assumptions 1 and 2 hold and consider the online setting of problem (P). Apply SpiderBoot-O with parameters $|S_1| = 24\sigma^2\epsilon^{-2}, q = |S| = \sqrt{|S_1|}, \eta = \frac{1}{2L}$. Then, the corresponding output $x_\xi$ satisfies $\mathbb{E}\|\nabla f(x_\xi)\| \leq \epsilon$ provided that the total number $K$ of iterations satisfies*

$$K \geq \mathcal{O}\Big(\frac{L(\Psi(x_0) - \Psi^*)}{\epsilon^2}\Big).$$

*Moreover, the resulting total SFO complexity is $\mathcal{O}(\epsilon^{-3} + \epsilon^{-2})$, and the PO complexity is $\mathcal{O}(\epsilon^{-2})$.*

### D.2 Constrained Optimization under Non-Euclidean Geometry

Algorithm 5 can be generalized to solve the online optimization problem (R) subject to a convex constraint set $\mathcal{X}$ with a general distance function. To do this, one replaces the proximal gradient update in Algorithm 5 with the generalized proximal gradient step in eq. (6) which is based on a proper Bregman distance $V$. For such an algorithm, we obtain the following result on the oracle complexity for Prox-SpiderBoost-O in solving constrained stochastic composite optimization under non-Euclidean geometry.

**Theorem 7.** *Let Assumptions 1 and 2 hold and consider the problem (R). Apply Prox-SpiderBoost-O with a proper Bregman distance $V$ that is $\alpha$-strongly convex with $\alpha > \frac{7}{8}$. Choose the parameters as $|S_1| = 2\left(\left(\alpha - \frac{7}{8}\right)^{-1}\left(1 + \frac{1}{4\alpha^2}\right) + \frac{2}{\alpha^2}\right)\sigma^2\epsilon^{-2}, \eta = \frac{1}{2L}$, and $q = |S| = \sqrt{|S_1|}$. Then, the algorithm outputs a point $x_\xi$ that satisfies $\mathbb{E}\|G_\eta(x_\xi)\| \leq \epsilon$ provided that the total number $K$ of iterations satisfies*

$$K \geq \frac{8L(\Psi(x_0) - \Psi^*)}{\epsilon^2}\left(\alpha - \frac{7}{8}\right)^{-1}\left(1 + \frac{1}{4\alpha}\right).$$

*Moreover, the overall SFO complexity is $\mathcal{O}(\epsilon^{-3} + \epsilon^{-2})$ and the PO complexity is $\mathcal{O}(\epsilon^{-2})$.*

## E   Prox-SpiderBoost-M-O for Online Nonconvex Composite Optimization

As the online problem (R) depends on the population risk that contains infinite samples, we propose a variant of Prox-SpiderBoost-M that can solve it in an online setting. We summarize the detailed steps of the algorithm in Algorithm 6, where we refer to it as Prox-SpiderBoost-M-O.

---

**Algorithm 6** Prox-SpiderBoost-M-O

---

**Input:** $q, K \in \mathbb{N}, \{\lambda_k\}_{k=1}^{K-1}, \{\beta_k\}_{k=1}^{K-1} > 0$.
**Set:** $\alpha_k = \frac{2}{\lceil k/q \rceil + 1}$.
**Initialize:** $y_0 = x_0 \in \mathbb{R}^d$.
**for** $k = 0, 1, \ldots, K-1$ **do**
  $z_k = (1 - \alpha_{k+1})y_k + \alpha_{k+1}x_k$,
  **if** $mod(k, q) = 0$ **then**
    draw $\xi_1$ data samples and compute $v_k = \frac{1}{|\xi_1|}\sum_{i=1}^{|\xi_1|}\nabla f_{u_i}(x)$
  **else**
    draw $\xi_2$ data samples and compute $v_k = \frac{1}{|\xi_2|}\sum_{i=1}^{|\xi_2|}(\nabla f_{u_i}(z_k) - \nabla f_{u_i}(z_{k-1})) + v_{k-1}$.
  **end**
  $x_{k+1} = \text{prox}_{\lambda_k g}(x_k - \lambda_k v_k)$,
  $y_{k+1} = z_k - \frac{\beta_k}{\lambda_k}x_k + \frac{\beta_k}{\lambda_k}\text{prox}_{\lambda_k g}(x_k - \lambda_k v_k)$.
**end**
**Output:** $z_\zeta$, where $\zeta \overset{\text{Unif}}{\sim} \{0, \ldots, K-1\}$.

---

Note that unlike the Prox-SpiderBoost-M for the finite-sum case, the Prox-SpiderBoost-M-O keeps drawing new data samples from the underlying distribution to construct the gradient estimate $v_k$. To study its convergence guarantee, we make the following standard assumption on the variance of the random sampling. We next present the convergence guarantee for Prox-SpiderBoost-M-O.

**Theorem 8.** *Let Assumptions 1 and 2 hold. Apply Prox-SpiderBoost-M-O (see Algorithm 6) to solve the problem (R). Choose any desired accuracy $\epsilon > 0$ and set parameters $\alpha_k = \frac{2}{k+1}, q = |\xi_2| = \sqrt{|\xi_1|} = \sqrt{\frac{2\sigma^2}{\epsilon^2}}, \beta_k \equiv \frac{1}{8L}$ and $\lambda_k \in [\beta_k, (1+\alpha_k)\beta_k]$. Then, the output $z_\zeta$ of the algorithm satisfies $\mathbb{E}\|G_{\lambda_\zeta}(z_\zeta, \nabla f(z_\zeta))\| \le \epsilon$ provided that the total number of iterations $K$ satisfies*

$$K \ge \Theta\left(\frac{L(F(x_0) - F^*)}{\epsilon^2}\right). \tag{7}$$

*Moreover, the total number of stochastic gradient calls is at most $\Theta(\epsilon^{-3})$ and the total number of proximal mapping calls is at most $\Theta(\epsilon^{-2})$.*

The orders of the results in Theorem 8 match those of state-of-arts [8]. Our result demonstrates that the momentum scheme can be applied to facilitate the convergence of Prox-SpiderBoost for solving online nonsmooth and nonconvex problems with a provable convergence guarantee.

## F   Objective Functions in Experiments

We specify the two objective functions that we adopt in our experiments. The nonsmooth problems are the regularized versions of these problems. The first problem is the logistic regression problem with a nonconvex regularizer, which takes the following form

$$\min_{w \in \mathbb{R}^d} f(w) := \frac{1}{n}\sum_{i=1}^{n}\ell(w^{\intercal}x_i, y_i) + \alpha\sum_{i=1}^{d}\frac{w_i^2}{1 + w_i^2},$$

where $x_i \in \mathbb{R}^d$ denotes the features and $y_i \in \{\pm 1\}$ corresponds to the labels, and $\alpha = 0.1$. We set the loss $\ell$ to be the cross-entropy loss given by

$$\ell(w^{\intercal}x_i, y_i) = -y_i \log\left(\frac{1}{1 + e^{-w^T x_i}}\right).$$

The second loss function is the following nonconvex robust linear regression problem

$$\min_{w \in \mathbb{R}^d} f(w) := \frac{1}{n}\sum_{i=1}^{n}\ell(y_i - w^{\intercal}x_i),$$

where we use the nonconvex loss function $\ell(x) := \log(\frac{x^2}{2} + 1)$.

# Technical Proofs

## G   Analysis of SpiderBoost (Proof of Theorem 1)

Throughout the paper, let $n_k = \lceil k/q \rceil$ such that $(n_k - 1)q \leq k \leq n_k q - 1$. Next, we establish our main result that yields Theorem 1.

**Theorem 9.** *Under Assumption 1, if the parameters $\eta, q$ and $S$ are chosen such that*

$$\beta_1 \triangleq \frac{\eta}{2} - \frac{L\eta^2}{2} - \frac{\eta^3 L^2 q}{2|S|} > 0, \tag{8}$$

*and if it holds that for $\mathrm{mod}(k, q) = 0$, we always have*

$$\mathbb{E}\|v_k - \nabla f(x_k)\|^2 \leq \epsilon_1^2, \tag{9}$$

*then the output point $x_\xi$ of SpiderBoost satisfies*

$$\mathbb{E}\|\nabla f(x_\xi)\|^2 \leq \frac{2}{K\beta_1}\left(1 + \frac{L^2\eta^2 q}{|S|}\right)(f(x_0) - f^*) + \left(\frac{\eta}{\beta_1} + 2 + \frac{L^2\eta^3 q}{|S|\beta_1}\right)\epsilon_1^2. \tag{10}$$

### G.1   Proof of Theorem 9

We first present an auxiliary lemma from [8].

**Lemma 1** ([8], Lemma 1). *Let Assumption 1 hold. The gradient estimator $v_k$ generated by eq. (1) satisfies for all $(n_k - 1)q + 1 \leq k \leq n_k q - 1$,*

$$\mathbb{E}\|v_k - \nabla f(x_k)\|^2 \leq \frac{L^2}{|S|}\mathbb{E}\|x_k - x_{k-1}\|^2 + \mathbb{E}\|v_{k-1} - \nabla f(x_{k-1})\|^2. \tag{11}$$

Telescoping Lemma 1 over $k$ from $(n_k - 1)q + 1$ to $k$, where $k \leq n_k q - 1$, we obtain that

$$\mathbb{E}\|v_k - \nabla f(x_k)\|^2 \leq \sum_{i=(n_k-1)q}^{k-1} \frac{L^2}{|S|}\mathbb{E}\|x_{i+1} - x_i\|^2 + \mathbb{E}\|v_{(n_k-1)q} - \nabla f(x_{(n_k-1)q})\|^2$$

$$\leq \sum_{i=(n_k-1)q}^{k} \frac{L^2}{|S|}\mathbb{E}\|x_{i+1} - x_i\|^2 + \mathbb{E}\|v_{(n_k-1)q} - \nabla f(x_{(n_k-1)q})\|^2. \tag{12}$$

We note that the above inequality also holds for $k = (n_k - 1)q$, which can be simply checked by plugging $k = (n_k - 1)q$ into above inequality.

*Proof.* By Assumption 1, the entire objective function $f$ is $L$-smooth, which further implies that

$$f(x_{k+1}) \leq f(x_k) + \langle \nabla f(x_k), x_{k+1} - x_k \rangle + \frac{L}{2}\|x_{k+1} - x_k\|^2$$

$$\overset{(i)}{=} f(x_k) - \eta\langle \nabla f(x_k), v_k \rangle + \frac{L\eta^2}{2}\|v_k\|^2$$

$$= f(x_k) - \eta\langle \nabla f(x_k) - v_k, v_k \rangle - \eta\|v_k\|^2 + \frac{L\eta^2}{2}\|v_k\|^2$$

$$\overset{(ii)}{\leq} f(x_k) + \frac{\eta}{2}\|\nabla f(x_k) - v_k\|^2 - (\frac{\eta}{2} - \frac{L\eta^2}{2})\|v_k\|^2,$$

where (i) follows from the update rule of SpiderBoost, (ii) uses the inequality that $\langle x, y \rangle \leq (\|x\|^2 + \|y\|^2)/2$ for all $x, y \in \mathbb{R}^d$. Taking expectation on both sides of the above inequality yields that

$$\mathbb{E}f(x_{k+1})$$

$$\leq \mathbb{E}f(x_k) + \frac{\eta}{2}\mathbb{E}\|\nabla f(x_k) - v_k\|^2 - (\frac{\eta}{2} - \frac{L\eta^2}{2})\mathbb{E}\|v_k\|^2$$

$$\overset{(i)}{\leq} \mathbb{E}f(x_k) + \frac{\eta}{2}\sum_{i=(n_k-1)q}^{k}\frac{L^2}{|S|}\mathbb{E}\|x_{i+1} - x_i\|^2 + \frac{\eta}{2}\mathbb{E}\|v_{(n_k-1)q} - \nabla f(x_{(n_k-1)q})\|^2 - (\frac{\eta}{2} - \frac{L\eta^2}{2})\mathbb{E}\|v_k\|^2$$

$$\overset{(ii)}{=} \mathbb{E}f(x_k) + \frac{\eta^3}{2}\sum_{i=(n_k-1)q}^{k}\frac{L^2}{|S|}\mathbb{E}\|v_i\|^2 + \frac{\eta}{2}\epsilon_1^2 - (\frac{\eta}{2} - \frac{L\eta^2}{2})\mathbb{E}\|v_k\|^2, \tag{13}$$

where (i) follows from eq. (12), and (ii) follows from eq. (9), and the fact that $x_{k+1} = x_k - \eta v_k$. Next, telescoping eq. (13) over $k$ from $(n_k - 1)q$ to $k$ where $k \leq n_k q - 1$ and noting that for $(n_k - 1)q \leq j \leq n_k q - 1$, $n_j = n_k$, we obtain

$$\mathbb{E}f(x_{k+1})$$

$$\leq \mathbb{E}f(x_{(n_k-1)q}) + \frac{\eta^3}{2}\sum_{j=(n_k-1)q}^{k}\sum_{i=(n_k-1)q}^{j}\frac{L^2}{|S|}\mathbb{E}\|v_i\|^2 + \frac{\eta}{2}\sum_{j=(n_k-1)q}^{k}\epsilon_1^2 - (\frac{\eta}{2} - \frac{L\eta^2}{2})\sum_{j=(n_k-1)q}^{k}\mathbb{E}\|v_j\|^2$$

$$\overset{(i)}{\leq} \mathbb{E}f(x_{(n_k-1)q}) + \frac{\eta^3}{2}\sum_{j=(n_k-1)q}^{k}\sum_{i=(n_k-1)q}^{k}\frac{L^2}{|S|}\mathbb{E}\|v_i\|^2 + \frac{\eta}{2}\sum_{j=(n_k-1)q}^{k}\epsilon_1^2 - (\frac{\eta}{2} - \frac{L\eta^2}{2})\sum_{j=(n_k-1)q}^{k}\mathbb{E}\|v_j\|^2$$

$$\overset{(ii)}{\leq} \mathbb{E}f(x_{(n_k-1)q}) + \frac{\eta^3 L^2 q}{2|S|}\sum_{i=(n_k-1)q}^{k}\mathbb{E}\|v_i\|^2 + \frac{\eta}{2}\sum_{j=(n_k-1)q}^{k}\epsilon_1^2 - (\frac{\eta}{2} - \frac{L\eta^2}{2})\sum_{j=(n_k-1)q}^{k}\mathbb{E}\|v_j\|^2$$

$$= \mathbb{E}f(x_{(n_k-1)q}) - \sum_{i=(n_k-1)q}^{k}\left(\frac{\eta}{2} - \frac{L\eta^2}{2} - \frac{\eta^3 L^2 q}{2|S|}\right)\mathbb{E}\|v_i\|^2 + \frac{\eta}{2}\sum_{i=(n_k-1)q}^{k}\epsilon_1^2$$

$$\overset{(iii)}{=} \mathbb{E}f(x_{(n_k-1)q}) - \sum_{i=(n_k-1)q}^{k}\left(\beta_1\mathbb{E}\|v_i\|^2 - \frac{\eta}{2}\epsilon_1^2\right) \tag{14}$$

where (i) extends the summation of the second term from $j$ to $k$, (ii) follows from the fact that $k \leqslant n_k q - 1$. Thus, we obtain

$$\sum_{j=(n_k-1)q}^{k}\sum_{i=(n_k-1)q}^{k}\frac{L^2}{|S|}\mathbb{E}\|v_i\|^2 \leq \frac{(k+q-n_kq+1)L^2}{|S|}\sum_{i=(n_k-1)q}^{k}\mathbb{E}\|v_i\|^2 \leq \frac{qL^2}{|S|}\sum_{i=(n_k-1)q}^{k}\mathbb{E}\|v_i\|^2,$$

and (iii) follows from the definition of $\beta_1$.

We continue the proof by further driving

$$\mathbb{E}f(x_K) - \mathbb{E}f(x_0)$$

$$= (\mathbb{E}f(x_q) - \mathbb{E}f(x_0)) + (\mathbb{E}f(x_{2q}) - \mathbb{E}f(x_q)) + \cdots + (\mathbb{E}f(x_K) - \mathbb{E}f(x_{(n_k-1)q}))$$

$$\overset{(i)}{\leq} -\sum_{i=0}^{q-1}\left(\beta_1\mathbb{E}\|v_i\|^2 - \frac{\eta}{2}\epsilon_1^2\right) - \sum_{i=q}^{2q-1}\left(\beta_1\mathbb{E}\|v_i\|^2 - \frac{\eta}{2}\epsilon_1^2\right) - \cdots - \sum_{i=(n_K-1)q}^{K-1}\left(\beta_1\mathbb{E}\|v_i\|^2 - \frac{\eta}{2}\epsilon_1^2\right)$$

$$= -\sum_{i=0}^{K-1}\left(\beta_1\mathbb{E}\|v_i\|^2 - \frac{\eta}{2}\epsilon_1^2\right) = -\sum_{i=0}^{K-1}\beta_1\mathbb{E}\|v_i\|^2 + \frac{K\eta}{2}\epsilon_1^2,$$

where (i) follows from eq. (14). Note that $\mathbb{E}f(x_K) \geq f^* \triangleq \inf_{x\in\mathbb{R}^d} f(x)$. Hence, the above inequality implies that

$$\sum_{i=0}^{K-1}\beta_1\mathbb{E}\|v_i\|^2 \leq f(x_0) - f^* + \frac{K\eta}{2}\epsilon_1^2. \tag{15}$$

We next bound $\mathbb{E}\|\nabla f(x_\xi)\|^2$, where $\xi$ is selected uniformly at random from $\{0, \ldots, K-1\}$. Observe that

$$\mathbb{E}\|\nabla f(x_\xi)\|^2 = \mathbb{E}\|\nabla f(x_\xi) - v_\xi + v_\xi\|^2 \leq 2\mathbb{E}\|\nabla f(x_\xi) - v_\xi\|^2 + 2\mathbb{E}\|v_\xi\|^2. \qquad (16)$$

Next, we bound the two terms on the right hand side of the above inequality. First, note that

$$\mathbb{E}\|v_\xi\|^2 = \frac{1}{K} \sum_{i=0}^{K-1} \mathbb{E}\|v_i\|^2 \leq \frac{f(x_0) - f^*}{K\beta_1} + \frac{\eta}{2\beta_1}\epsilon_1^2, \qquad (17)$$

where the last inequality follows from eq. (15). On the other hand, note that

$$\mathbb{E}\|\nabla f(x_\xi) - v_\xi\|^2 \overset{(i)}{\leq} \mathbb{E} \sum_{i=(n_\xi-1)q}^{\xi} \frac{L^2}{|S|} \mathbb{E}\|x_{i+1} - x_i\|^2 + \epsilon_1^2 \overset{(ii)}{=} \epsilon_1^2 + \mathbb{E} \sum_{i=(n_\xi-1)q}^{\xi} \frac{L^2\eta^2}{|S|} \mathbb{E}\|v_i\|^2$$

$$\overset{(iii)}{\leq} \epsilon_1^2 + \mathbb{E} \sum_{i=(n_\xi-1)q}^{\min\{(n_\xi)q-1, K-1\}} \frac{L^2\eta^2}{|S|} \mathbb{E}\|v_i\|^2 \overset{(iv)}{\leq} \epsilon_1^2 + \frac{q}{K} \sum_{i=0}^{K-1} \frac{L^2\eta^2}{|S|} \mathbb{E}\|v_i\|^2$$

$$\overset{(v)}{\leq} \epsilon_1^2 + \frac{L^2\eta^2 q}{K|S|\beta_1} (f(x_0) - f^*) + \frac{L^2\eta^3 q}{2|S|\beta_1}\epsilon_1^2, \qquad (18)$$

where (i) follows from eqs. (9) and (12), (ii) follows from the fact that $x_{k+1} = x_k - \eta v_k$, (iii) follows from the definition of $n_\xi$, which implies $\xi \leqslant \min\{(n_\xi)q-1, K-1\}$, (iv) follows from the fact that the probability that $n_\xi = 1, 2, \cdots, n_K$ is less than or equal to $q/(K)$, and (v) follows from eq. (15).

Substituting eqs. (17) and (18) into eq. (16), we obtain

$$\mathbb{E}\|\nabla f(x_\xi)\|^2 \leq \frac{2(f(x_0) - f^*)}{K\beta_1} + \frac{\eta}{\beta_1}\epsilon_1^2 + 2\epsilon_1^2 + \frac{2L^2\eta^2 q}{K|S|\beta_1} (f(x_0) - f^*) + \frac{L^2\eta^3 q}{|S|\beta_1}\epsilon_1^2$$

$$= \frac{2}{K\beta_1}\left(1 + \frac{L^2\eta^2 q}{|S|}\right)(f(x_0) - f^*) + \left(\frac{\eta}{\beta_1} + 2 + \frac{L^2\eta^3 q}{|S|\beta_1}\right)\epsilon_1^2.$$

$\square$

## G.2  Proof of Theorem 1

Based on the parameter setting in Theorem 1 that

$$q = \sqrt{n}, S = \sqrt{n}, \text{ and } \eta = \frac{1}{2L}, \qquad (19)$$

we obtain

$$\beta_1 = \frac{\eta}{2} - \frac{L\eta^2}{2} - \frac{\eta^3 L^2 q}{2|S|} = \frac{1}{16L} > 0. \qquad (20)$$

Moreover, for $\text{mod}(k, q) = 0$, as the algorithm is designed to take the full-batch gradient of the finite-sum problem, we have

$$\mathbb{E}\|v_k - \nabla f(x_k)\|^2 = \mathbb{E}\|\nabla f(x_k) - \nabla f(x_k)\|^2 = 0. \qquad (21)$$

Equations (20) and (21) imply that the parameters in Theorem 1 satisfy the assumptions in Theorem 9 with $\beta_1 = 1/(16L)$ and $\epsilon_1 = 0$. Plugging eqs. (19) to (21) into Theorem 9, we obtain that, after $K$ iterations, the output of SpiderBoost satisfies

$$\mathbb{E}\|\nabla f(x_\xi)\|^2 \leq \frac{40L}{K}(f(x_0) - f^*). \qquad (22)$$

To ensure $\mathbb{E}\|\nabla f(x_\xi)\| \leqslant \epsilon$, it is sufficient to ensure $\mathbb{E}\|\nabla f(x_\xi)\|^2 \leqslant \epsilon^2$ (because $(\mathbb{E}\|\nabla f(x_\xi)\|)^2 \leq \mathbb{E}\|\nabla f(x_\xi)\|^2$ due to Jensen's inequality). Thus, we need the total number $K$ of iterations satisfies that $\frac{40L}{K}(f(x_0) - f^*) \leq \epsilon^2$, which gives

$$K = \frac{40L}{\epsilon^2}(f(x_0) - f^*). \qquad (23)$$

Then, the total SFO complexity is given by

$$\left\lceil \frac{K}{q} \right\rceil \cdot n + K \cdot S \leqslant (K + q) \cdot \frac{n}{q} + K \cdot S = K\sqrt{n} + n + K\sqrt{n} = \mathcal{O}(\sqrt{n}\epsilon^{-2} + n),$$

where the last equation follows from eq. (23), which completes the proof.

# H  Analysis of Prox-SpiderBoost and Prox-SpiderBoost-O (Proofs of Theorems 2, 4, 6 and 7)

We first establish the following major theorem, which is applicable to both the finite-sum and the online problem. We then generalize it for these two cases.

**Theorem 10.** *Under Assumption 1, choose a proper prox-funtion $V(\cdot) : \mathcal{X} \to \mathbb{R}$ with modulus $\alpha$. Then, if the parameters $\eta, q$ and $S$ are chosen such that*

$$\beta_2 \triangleq \alpha\eta - \frac{L\eta^2}{2} - \frac{\eta}{2} - \frac{\eta^3 L^2 q}{2|S|} > 0, \tag{24}$$

*and if it holds that for $\mathrm{mod}(k, q) = 0$, we always have*

$$\mathbb{E}\|v_k - \nabla f(x_k)\|^2 \leq \epsilon_1^2, \tag{25}$$

*then, the output point $x_\xi$ of Prox-SpiderBoost or Prox-SpiderBoost-O satisfies*

$$\mathbb{E}\|\tilde{g}_\xi\|^2 \leq \frac{2}{K\beta_2}\left(1 + \frac{L^2\eta^2 q}{\alpha^2|S|}\right)(f(x_0) - f^*) + \left(\frac{\eta}{\beta_2} + \frac{2}{\alpha^2} + \frac{L^2\eta^3 q}{\alpha^2|S|\beta_2}\right)\epsilon_1^2, \tag{26}$$

*where $\tilde{g}_\xi = P_\mathcal{X}(x_\xi, \nabla f(x_\xi), \eta)$.*

As stated in the theorem, we require $\beta = \left(\alpha\eta - \frac{L\eta^2}{2} - \frac{\eta}{2} - \frac{\eta^3 L^2}{2}\right) > 0$ to conclude our theorem. A simple case would be $\eta = 1/(2L)$ and $w(x) = \|x\|^2/2$, which gives $\alpha = 1$ and $\beta = 1/(16L)$.

## H.1  Proof of Theorem 10

To prove Theorem 10, we first introduce a useful lemma.

**Lemma 2** ([11], Lemma 1 and Proposition 1). *Let $\mathcal{X}$ be a closed convex set in $\mathbb{R}^d$, $h : \mathcal{X} \to \mathbb{R}$ be a convex function, but possibly nonsmooth, and $V : \mathcal{X} \to \mathbb{R}$ be defined in eq. (5). Moreover, define*

$$x^+ = \arg\min_{u \in \mathcal{X}}\left\{\langle g, u\rangle + \frac{1}{\eta}V(u, x) + h(u)\right\} \tag{27}$$

$$P_\mathcal{X}(x, g, \eta) = \frac{1}{\eta}(x - x^+), \tag{28}$$

*where $g \in \mathbb{R}^d$, $x \in \mathcal{X}$, and $\eta > 0$. Then, the following statement hold*

$$\langle g, P_\mathcal{X}(x, g, \eta)\rangle \geqslant \alpha\|P_\mathcal{X}(x, g, \eta)\|^2 + \frac{1}{\eta}[h(x^+) - h(x)]. \tag{29}$$

*Moreover, for any $g_1, g_2 \in \mathbb{R}^d$, we have*

$$\|P_\mathcal{X}(x, g_1, \eta) - P_\mathcal{X}(x, g_2, \eta)\| \leqslant \frac{1}{\alpha}\|g_1 - g_2\|. \tag{30}$$

Now, we are ready to prove Theorem 10. To ease our nation, let $g_k = P_\mathcal{X}(x_k, v_k, \eta)$, which is defined in eq. (28). We begin with the analysis at iteration $k$. By the Lipschitz continuity of $\nabla f$, we obtain

$$
\begin{aligned}
f(x_{k+1}) &\leq f(x_k) + \langle\nabla f(x_k), x_{k+1} - x_k\rangle + \frac{L}{2}\|x_{k+1} - x_k\|^2 \\
&\overset{(i)}{=} f(x_k) - \eta\langle\nabla f(x_k), g_k\rangle + \frac{L\eta^2}{2}\|g_k\|^2 \\
&= f(x_k) - \eta\langle\nabla f(x_k) - v_k, g_k\rangle - \eta\langle v_k, g_k\rangle + \frac{L\eta^2}{2}\|g_k\|^2 \\
&\overset{(ii)}{\leq} f(x_k) + \frac{\eta}{2}\|\nabla f(x_k) - v_k\|^2 - \eta\langle v_k, g_k\rangle + \left(\frac{L\eta^2}{2} + \frac{\eta}{2}\right)\|g_k\|^2 \\
&\overset{(iii)}{\leq} f(x_k) + \frac{\eta}{2}\|\nabla f(x_k) - v_k\|^2 - \alpha\eta\|g_k\|^2 + h(x_k) - h(x_{k+1}) + \left(\frac{L\eta^2}{2} + \frac{\eta}{2}\right)\|g_k\|^2
\end{aligned}
\tag{31}
$$

where (i) follows from the definition of $P_{\mathcal{X}}(x_k, v_k, \eta)$, and the update rule of Prox-SpiderBoost and Prox-SpiderBoost-O, (ii) follows from the inequality that $\langle x, y \rangle \leq \frac{\|x\|^2 + \|y\|^2}{2}$ for $x, y \in \mathbb{R}^d$, and (iii) follows from eq. (29) with $g = v_k, x = x_k$ and $P_{\mathcal{X}}(x_k, v_k, \eta) = g_k$.

Taking expectation on both sides of eq. (31), and arranging it with the definition of $\Psi(x) := f(x) + h(x)$, we obtain

$$
\mathbb{E}\Psi(x_{k+1}) \leq \mathbb{E}\Psi(x_k) + \frac{\eta}{2}\mathbb{E}\|\nabla f(x_k) - v_k\|^2 - \left(\alpha\eta - \frac{L\eta^2}{2} - \frac{\eta}{2}\right)\mathbb{E}\|g_k\|^2
$$

$$
\overset{(i)}{\leq} \mathbb{E}\Psi(x_k) + \frac{\eta}{2}\sum_{i=(n_k-1)q}^{k}\frac{L^2}{|S|}\mathbb{E}\|x_{i+1} - x_i\|^2
$$

$$
+ \frac{\eta}{2}\mathbb{E}\|v_{(n_k-1)q} - \nabla f(x_{(n_k-1)q})\|^2 - \left(\alpha\eta - \frac{L\eta^2}{2} - \frac{\eta}{2}\right)\mathbb{E}\|g_k\|^2
$$

$$
\overset{(ii)}{\leq} \mathbb{E}\Psi(x_k) + \frac{\eta^3}{2}\sum_{i=(n_k-1)q}^{k}\frac{L^2}{|S|}\mathbb{E}\|g_i\|^2 + \frac{\eta\epsilon_1^2}{2} - \left(\alpha\eta - \frac{L\eta^2}{2} - \frac{\eta}{2}\right)\mathbb{E}\|g_k\|^2
$$

where (i) follows from eq. (12), and (ii) follows from eq. (25) and the fact that $x_{k+1} = x_k - \eta g_k$. Telescoping the above inequality over $k$ from $(n_k - 1)q$ to $k$ where $k \leq n_k q - 1$ and noting that for $(n_k - 1)q \leq j \leq n_k q - 1$, $n_j = n_k$, we have

$$
\mathbb{E}\Psi(x_{k+1}) - \mathbb{E}\Psi(x_{(n_k-1)q})
$$

$$
\leq \frac{\eta^3}{2}\sum_{j=(n_k-1)q}^{k}\sum_{i=(n_k-1)q}^{j}\frac{L^2}{|S|}\mathbb{E}\|g_i\|^2 + \frac{\eta}{2}\sum_{j=(n_k-1)q}^{k}\epsilon_1^2 - \left(\alpha\eta - \frac{L\eta^2}{2} - \frac{\eta}{2}\right)\sum_{j=(n_k-1)q}^{k}\mathbb{E}\|g_j\|^2
$$

$$
\overset{(i)}{\leq} \frac{\eta^3}{2}\sum_{j=(n_k-1)q}^{k}\sum_{i=(n_k-1)q}^{k}\frac{L^2}{|S|}\mathbb{E}\|g_i\|^2 + \frac{\eta}{2}\sum_{j=(n_k-1)q}^{k}\epsilon_1^2 - \left(\alpha\eta - \frac{L\eta^2}{2} - \frac{\eta}{2}\right)\sum_{j=(n_k-1)q}^{k}\mathbb{E}\|g_j\|^2
$$

$$
\overset{(ii)}{\leq} \frac{\eta^3 L^2 q}{2|S|}\sum_{i=(n_k-1)q}^{k}\mathbb{E}\|g_i\|^2 + \frac{\eta}{2}\sum_{j=(n_k-1)q}^{k}\epsilon_1^2 - \left(\alpha\eta - \frac{L\eta^2}{2} - \frac{\eta}{2}\right)\sum_{j=(n_k-1)q}^{k}\mathbb{E}\|g_j\|^2
$$

$$
= -\sum_{i=(n_k-1)q}^{k}\left(\alpha\eta - \frac{L\eta^2}{2} - \frac{\eta}{2} - \frac{\eta^3 L^2 q}{2|S|}\right)\mathbb{E}\|g_i\|^2 + \frac{\eta}{2}\sum_{i=(n_k-1)q}^{k}\epsilon_1^2
$$

$$
\overset{(iii)}{=} -\sum_{i=(n_k-1)q}^{k}\left(\beta_2\mathbb{E}\|g_i\|^2 - \frac{\eta}{2}\epsilon_1^2\right) \tag{32}
$$

where (i) extends the summation of second term from $j$ to $k$, (ii) follows from the fact that $k \leqslant n_k q - 1$ and thus

$$
\sum_{j=(n_k-1)q}^{k}\sum_{i=(n_k-1)q}^{k}\frac{L^2}{|S|}\mathbb{E}\|g_i\|^2 \leq \frac{(k + q - n_k q + 1)L^2}{|S|}\sum_{i=(n_k-1)q}^{k}\mathbb{E}\|g_i\|^2 \leq \frac{qL^2}{|S|}\sum_{i=(n_k-1)q}^{k}\mathbb{E}\|g_i\|^2,
$$

and (iii) follows from the definition of $\beta_2$. We continue to derive

$$
\mathbb{E}\Psi(x_K) - \mathbb{E}\Psi(x_0)
$$

$$
= (\mathbb{E}\Psi(x_q) - \mathbb{E}\Psi(x_0)) + (\mathbb{E}\Psi(x_{2q}) - \mathbb{E}\Psi(x_q)) + \cdots + (\mathbb{E}\Psi(x_K) - \mathbb{E}\Psi(x_{(n_k-1)q}))
$$

$$
\overset{(i)}{\leq} -\sum_{i=0}^{q-1}\left(\beta_2\mathbb{E}\|g_i\|^2 - \frac{\eta}{2}\epsilon_1^2\right) - \sum_{i=q}^{2q-1}\left(\beta_2\mathbb{E}\|g_i\|^2 - \frac{\eta}{2}\epsilon_1^2\right) - \cdots - \sum_{i=(n_K-1)q}^{K-1}\left(\beta_2\mathbb{E}\|g_i\|^2 - \frac{\eta}{2}\epsilon_1^2\right)
$$

$$
= -\sum_{i=0}^{K-1}\left(\beta_2\mathbb{E}\|g_i\|^2 - \frac{\eta}{2}\epsilon_1^2\right) = -\sum_{i=0}^{K-1}\beta_2\mathbb{E}\|g_i\|^2 + \frac{K\eta}{2}\epsilon_1^2,
$$

where (i) follows from eq. (32). Note that $\mathbb{E}\Psi(x_K) \geq \Psi^* \triangleq \inf_{x \in \mathbb{R}^d} \Psi(x)$. The above inequality implies that

$$\sum_{i=0}^{K-1} \beta_2 \mathbb{E}\|g_i\|^2 \leq \Psi(x_0) - \Psi^* + \frac{K\eta}{2}\epsilon_1^2. \tag{33}$$

We next bound the output of algorithms. Define $\tilde{g}_\xi = P(x_\xi, \nabla f(x_\xi), \eta)$, where $\xi$ is selected uniformly at random from $\{0, \ldots, K-1\}$. Observe that

$$\mathbb{E}\|\tilde{g}_\xi\|^2 \leq 2\mathbb{E}\|g_\xi\|^2 + 2\mathbb{E}\|\tilde{g}_\xi - g_\xi\|^2 \overset{(i)}{\leq} 2\mathbb{E}\|g_\xi\|^2 + \frac{2}{\alpha^2}\mathbb{E}\|\nabla f(x_\xi) - v_\xi\|^2 \tag{34}$$

where (i) follows from the definition of $\tilde{g}_k, g_k$ and the property of $g_k$ and $\tilde{g}_k$ in eq. (30).

Next, we bound the two terms on the right hand side of the above inequality. First, note that

$$\mathbb{E}\|g_\xi\|^2 = \frac{1}{K}\sum_{i=0}^{K-1}\mathbb{E}\|g_i\|^2 \leq \frac{\Psi(x_0) - \Psi^*}{K\beta_2} + \frac{\eta}{2\beta_2}\epsilon_1^2, \tag{35}$$

where the last inequality follows from eq. (33). On the other hand, note that

$$\mathbb{E}\|\nabla f(x_\xi) - v_\xi\|^2 \overset{(i)}{\leq} \mathbb{E}\sum_{i=(n_\xi-1)q}^{\xi} \frac{L^2}{|S|}\mathbb{E}\|x_{i+1} - x_i\|^2 + \epsilon_1^2 \overset{(ii)}{=} \epsilon_1^2 + \mathbb{E}\sum_{i=(n_\xi-1)q}^{\xi} \frac{L^2\eta^2}{|S|}\mathbb{E}\|g_i\|^2$$

$$\overset{(iii)}{\leq} \epsilon_1^2 + \mathbb{E}\sum_{i=(n_\xi-1)q}^{\min\{(n_\xi)q-1,K-1\}} \frac{L^2\eta^2}{|S|}\mathbb{E}\|g_i\|^2 \overset{(iv)}{\leq} \epsilon_1^2 + \frac{q}{K}\sum_{i=0}^{K-1} \frac{L^2\eta^2}{|S|}\mathbb{E}\|g_i\|^2$$

$$\overset{(v)}{\leq} \epsilon_1^2 + \frac{L^2\eta^2 q}{K|S|\beta_2}\left(\Psi(x_0) - \Psi^*\right) + \frac{L^2\eta^3 q}{2|S|\beta_2}\epsilon_1^2, \tag{36}$$

where (i) follows from eqs. (12) and (25), (ii) follows from the fact that $x_{k+1} = x_k - \eta g_k$, (iii) follows from the definition of $n_\xi$, which implies $\xi \leqslant \min\{(n_\xi)q-1, K-1\}$, (iv) follows from the fact that the probability that $n_\xi = 1$ or $2, \cdots$ or $n_k$ is less than or equal to $q/K$, and (v) follows from eq. (35).

Substituting eqs. (35) and (36) into eq. (34) yields

$$\mathbb{E}\|\tilde{g}_\xi\|^2 \leq \frac{2\left(f(x_0) - f^*\right)}{K\beta_2} + \frac{\eta}{\beta_2}\epsilon_1^2 + \frac{2}{\alpha^2}\left(\epsilon_1^2 + \frac{L^2\eta^2 q}{K|S|\beta_2}\left(f(x_0) - f^*\right) + \frac{L^2\eta^3 q}{2|S|\beta_2}\epsilon_1^2\right)$$

$$= \frac{2}{K\beta_2}\left(1 + \frac{L^2\eta^2 q}{\alpha^2|S|}\right)\left(\Psi(x_0) - \Psi^*\right) + \left(\frac{\eta}{\beta_2} + \frac{2}{\alpha^2} + \frac{L^2\eta^3 q}{\alpha^2|S|\beta_2}\right)\epsilon_1^2,$$

which completes the proof.

## H.2 Proof of Theorem 2

*Proof.* Theorem 2 as a special case follows from the more general Theorem 4 that we develop in Appendix B with the choices of the Bregman distance function $V(x,y) = \frac{1}{2}\|x - y\|^2$ and $\alpha = 1$. □

## H.3 Proof of Theorem 4

Based on the parameter setting in Theorem 2 that

$$\alpha > \frac{7}{8}, q = \sqrt{n}, S = \sqrt{n}, \text{ and } \eta = \frac{1}{2L}, \tag{37}$$

we obtain

$$\beta_1 = \alpha\eta - \frac{L\eta^2}{2} - \frac{\eta}{2} - \frac{\eta^3 L^2 q}{2|S|} = \frac{1}{2L}\left(\alpha - \frac{7}{8}\right) > 0. \tag{38}$$

Moreover, for $\text{mod}(k, q) = 0$, as the algorithm is designed to take the full-batch gradient of the finite-sum problem, we have

$$\mathbb{E}\|v_k - \nabla f(x_k)\|^2 = \mathbb{E}\|\nabla f(x_k) - \nabla f(x_k)\|^2 = 0. \tag{39}$$

Equations (38) and (39) imply that the parameters in the finite-sum case satisfy the assumptions in Theorem 10 with $\beta_1 = (\alpha - 7/8)/(2L)$ and $\epsilon_1 = 0$. Plugging eqs. (37) to (39) into Theorem 10, we obtain that, after $K$ iterations, the output of Prox-SpiderBoost satisfies

$$\mathbb{E}\|\tilde{g}_\xi\|^2 \leq \frac{4L}{K}\left(\alpha - \frac{7}{8}\right)^{-1}\left(1 + \frac{1}{4\alpha}\right)(\Psi(x_0) - \Psi^*). \tag{40}$$

To ensure $\mathbb{E}\|\tilde{g}_\xi\| \leqslant \epsilon$, it is sufficient to ensure $\mathbb{E}\|\tilde{g}_\xi\|^2 \leqslant \epsilon^2$, thus, we obtain

$$K = \frac{4L}{\epsilon^2}\left(\alpha - \frac{7}{8}\right)^{-1}\left(1 + \frac{1}{4\alpha}\right)(\Psi(x_0) - \Psi^*). \tag{41}$$

Then, the SFO is

$$\left\lceil\frac{K}{q}\right\rceil \cdot n + K \cdot S \leqslant (K + q) \cdot \frac{n}{q} + K \cdot S = K\sqrt{n} + n + K\sqrt{n} = \mathcal{O}(\sqrt{n}\epsilon^{-2} + n),$$

where the last equation follows from eq. (23). The proximal oracle follows from the total iteration in eq. (41), which completes the proof.

### H.4 Proof of Theorem 6

*Proof.* Theorem 6 follows as a special case from the more general Theorem 7 that we develop in appendix D.2 with the choices of Bregman distance function $V(x, y) = \frac{1}{2}\|x - y\|^2$ and $\alpha = 1$. $\square$

### H.5 Proof of Theorem 7

Based on the parameter setting in Theorem 7 that

$$\alpha > \frac{7}{8}, S_1 = 2\left(\left(\alpha - \frac{7}{8}\right)^{-1}\left(1 + \frac{1}{4\alpha^2}\right) + \frac{2}{\alpha^2}\right)\sigma^2\epsilon^{-2}, q = \sqrt{S_1}, S = \sqrt{S_1}, \text{ and } \eta = \frac{1}{2L}, \tag{42}$$

we obtain

$$\beta_2 = \alpha\eta - \frac{L\eta^2}{2} - \frac{\eta}{2} - \frac{\eta^3 L^2 q}{2|S|} = \frac{1}{2L}\left(\alpha - \frac{7}{8}\right) > 0. \tag{43}$$

Moreover, for $\text{mod}(k, q) = 0$, we have

$$\mathbb{E}\|v_k - \nabla f(x_k)\|^2 = \mathbb{E}\left\|\frac{1}{|S_1|}\sum_{i \in S_1}\nabla f_i(x_k) - \nabla f(x_k)\right\|^2 = \frac{1}{|S_1|^2}\left\|\sum_{i \in S_1}\nabla f_i(x_k) - \nabla f(x_k)\right\|^2 \tag{44}$$

$$\overset{(i)}{=} \frac{1}{|S_1|^2}\sum_{i \in S_1}\|\nabla f_i(x_k) - \nabla f(x_k)\|^2 = \frac{1}{|S_1|}\|\nabla f_i(x_k) - \nabla f(x_k)\|^2 \overset{(ii)}{=} \frac{\sigma^2}{|S_1|} \tag{45}$$

$$\overset{(iii)}{\leq} \left(\left(\alpha - \frac{7}{8}\right)^{-1}\left(1 + \frac{1}{4\alpha^2}\right) + \frac{2}{\alpha^2}\right)^{-1}\frac{\epsilon^2}{2}. \tag{46}$$

where (i) follows from $\mathbb{E}\nabla f_i(x_k) - \nabla f(x_k) = 0$, and the fact that the samples from $S_1$ are drawn with replacement, and (iii) follows from eq. (42).

Equations (43) and (46) imply that the parameters in the online case satisfy the assumptions in Theorem 10 with $\beta_2 = (\alpha - 7/8)/(2L)$ and $\epsilon_1^2 = \left(\left(\alpha - \frac{7}{8}\right)^{-1}\left(1 + \frac{1}{4\alpha^2}\right) + \frac{2}{\alpha^2}\right)^{-1}\frac{\epsilon^2}{2}$. Plugging

eqs. (42), (43) and (46) into Theorem 10, we obtain that, after $K$ iterations, the output of Prox-SpiderBoost-O satisfies

$$\mathbb{E}\|\tilde{g}_\xi\|^2 \leq \frac{2}{K\beta_2}\left(1 + \frac{L^2\eta^2 q}{\alpha^2|S|}\right)(\Psi(x_0) - \Psi^*) + \left(\frac{\eta}{\beta_2} + \frac{2}{\alpha^2} + \frac{L^2\eta^3 q}{\alpha^2|S|\beta_2}\right)\epsilon_1^2 \qquad (47)$$

$$= \frac{4L}{K}\left(\alpha - \frac{7}{8}\right)^{-1}\left(1 + \frac{1}{4\alpha}\right)(\Psi(x_0) - \Psi^*) + \frac{\epsilon^2}{2}. \qquad (48)$$

To ensure $\mathbb{E}\|\tilde{g}_\xi\| \leqslant \epsilon$, it is sufficient to ensure $\mathbb{E}\|\tilde{g}_\xi\|^2 \leqslant \epsilon^2$, thus, we need

$$K = \frac{8L}{\epsilon^2}\left(\alpha - \frac{7}{8}\right)^{-1}\left(1 + \frac{1}{4\alpha}\right)(\Psi(x_0) - \Psi^*). \qquad (49)$$

Then, the total SFO complexity is

$$\left\lceil\frac{K}{q}\right\rceil \cdot S_1 + K \cdot S \leqslant (K + q) \cdot \frac{S_1}{q} + K \cdot S = K\sqrt{S_1} + S_1 + K\sqrt{S_1} = \mathcal{O}(\epsilon^{-3} + \epsilon^{-2}),$$

where the last equation follows from eq. (49). The proximal oracle follows from the total iteration in eq. (49), which finishes the proof.

### H.6 Proof of Corollary 1

*Proof.* Corollary 1 follows directly from Theorem 6, becasue the online setting of problem (P) is a special case of the problem (R). $\qquad \square$

## I Analysis of Prox-SpiderBoost-PL (Proof of Theorem 5)

Let us consider one outer loop. Following a similar proof as that of eq.(25) in [22], we obtain the following inequality for Prox-SpiderBoost-PL in finite-sum case.

$$\mathbb{E}\Psi(x_{k+1}) \leq \mathbb{E}\left[\Psi(x_k) - (\frac{1}{2\eta} - \frac{L}{2})\|x_{k+1} - x_k\|^2 - (\frac{1}{3\eta} - L)\|\overline{x_{k+1}} - x_k\|^2 + \eta\|\nabla f(x_k) - v_k\|^2\right],$$

where $\overline{x_{k+1}} := \text{prox}_{\eta g}(x_k - \eta\nabla f(x_k))$. Substituting the variance bound of Spider into the above inequality we obtain that

$$\mathbb{E}\Psi(x_{k+1}) \leq \mathbb{E}\left[\Psi(x_k) - (\frac{1}{2\eta} - \frac{L}{2})\|x_{k+1} - x_k\|^2 - (\frac{1}{3\eta} - L)\|\overline{x_{k+1}} - x_k\|^2 + \eta\sum_{i=(n_k-1)q}^{k}\frac{L^2}{|S|}\|x_{i+1} - x_i\|^2\right].$$

Summing the above inequality over $k$ from $(n_k - 1)q$ to $n_k q - 2$ and relax the upper bound of $i$ to $n_k q - 2$, we further obtain that

$$\mathbb{E}\Psi(x_{n_k q-1}) \leq \mathbb{E}\Psi(x_{(n_k-1)q}) - \sum_{i=(n_k-1)q}^{n_k q-2}(\frac{1}{2\eta} - \frac{L}{2} - \frac{\eta L^2(q-2)}{|S|})\mathbb{E}\|x_{i+1} - x_i\|^2 - (\frac{1}{3\eta} - L)\sum_{i=(n_k-1)q}^{n_k q-2}\mathbb{E}\|\overline{x_{i+1}} - x_i\|^2.$$

Noting that $q = |S|, \eta L = \frac{1}{8}$, we further obtain that

$$\mathbb{E}\Psi(x_{n_k q-1}) \leq \mathbb{E}\Psi(x_{(n_k-1)q}) - \sum_{i=(n_k-1)q}^{n_k q-2} 3L\mathbb{E}\|x_{i+1} - x_i\|^2 - L\eta^2\sum_{i=(n_k-1)q}^{n_k q-2}\mathbb{E}\|\mathcal{G}_\eta(x_i)\|^2$$

$$\leq \mathbb{E}\Psi(x_{(n_k-1)q}) - L\eta^2\sum_{i=(n_k-1)q}^{n_k q-2}\mathbb{E}\|\mathcal{G}_\eta(x_i)\|^2$$

Since $\mathbb{E}\Psi(x_{n_k q-1}) \geq \Psi^*$, the above inequality further implies that

$$\sum_{i=(n_k-1)q}^{n_k q-2}\mathbb{E}\|\mathcal{G}_\eta(x_i)\|^2 \leq 64L(\mathbb{E}\Psi(x_{(n_k-1)q}) - \Psi^*).$$

By the scheme of Prox-SpiderBoost-gd, we know that $\mathbb{E}\|\mathcal{G}_\eta(x_{n_k q})\|^2 = \frac{1}{q-2}\sum_{i=(n_k-1)q+1}^{n_k q-2}\mathbb{E}\|\mathcal{G}_\eta(x_i)\|^2$. Therefore, combining this inequality with the above inequality, we obtain that

$$\mathbb{E}\|\mathcal{G}_\eta(x_{n_k q})\|^2 = \frac{1}{q-2}\sum_{i=(n_k-1)q+1}^{n_k q-2}\mathbb{E}\|\mathcal{G}_\eta(x_i)\|^2$$
$$\leq \frac{64L}{q-2}\big(\mathbb{E}\Psi(x_{(n_k-1)q}) - \Psi^*\big)$$
$$\leq \frac{64L\tau}{q-2}\mathbb{E}\|\mathcal{G}_\eta(x_{(n_k-1)q})\|^2.$$

In order to produce a point such that $\mathbb{E}\|\mathcal{G}_\eta(x_{tq})\| \leq \epsilon$, we deduce from the above inequality that at least $t = \Theta(\log\frac{1}{\epsilon}/\log\frac{q}{L\tau})$ number of outer loops is needed. Note that $|S| = q = \Theta(L\tau)$, we conclude that $t = \Theta(\log\frac{1}{\epsilon})$. In summary, the total proximal oracle complexity (PO) is in the order $O(q\log\frac{1}{\epsilon}) = O(\tau\log\frac{1}{\epsilon})$, and the total stochastic first-order oracle complexity (SFO) is $O((n+q|S|)\log\frac{1}{\epsilon}) = O((n+L^2\tau^2)\log\frac{1}{\epsilon})$.

## J  Analysis of Prox-SpiderBoost-M and SpiderBoost-M-O (Proof of Theorem 3 and Theorem 8)

### J.1  Proof of Theorem 3

In this section, we provide the convergence analysis of Prox-SpiderBoost-M. Throughout, for any $k \in \mathbb{N}$, denote $\tau(k) \in \mathbb{N}$ the unique integer such that $(\tau(k)-1)q \leq k \leq \tau(k)q - 1$. We also define $\Gamma_0 = 0, \Gamma_1 = 1$ and $\Gamma_k = (1-\alpha_k)\Gamma_{k-1}$ for $k = 2, 3, \dots$. Since we set $\alpha_k = \frac{2}{\lceil k/q\rceil+1}$, it is easy to check that $\Gamma_k = \frac{2}{\lceil k/q\rceil(\lceil k/q\rceil+1)}$. We first provide some auxiliary lemmas that are useful for the analysis later.

**Auxiliary Lemmas**

We first present an auxiliary lemma from [8].

**Lemma 3.** *[8] Under Assumption 1, the estimation $v_k$ of gradient constructed by SPIDER satisfies that for all $(\tau(k)-1)q + 1 \leq k \leq \tau(k)q - 1$,*

$$\mathbb{E}\|v_k - \nabla f(z_k)\|^2 \leq \frac{L^2}{|\xi_k|}\mathbb{E}\|z_k - z_{k-1}\|^2 + \mathbb{E}\|v_{k-1} - \nabla f(z_{k-1})\|^2.$$

Telescoping Lemma 3 and noting that $v_k = \nabla f(z_k)$ for all $k$ such that $\mathrm{mod}(k,q) = 0$, we obtain the following bound.

**Lemma 4.** *Under Assumption 1, the estimation $v_k$ of gradient constructed by SPIDER satisfies that for all $k \in \mathbb{N}$,*

$$\mathbb{E}\|v_k - \nabla f(z_k)\|^2 \leq \sum_{i=(\tau(k)-1)q}^{k-1}\frac{L^2}{|\xi_i|}\mathbb{E}\|z_{i+1} - z_i\|^2. \tag{50}$$

Next, recall the following definition of the gradient mapping for some $\eta > 0$ and $x, u \in \mathbb{R}^d$:

$$G_\eta(x,u) := \frac{1}{\eta}\big(x - \mathrm{prox}_{\eta h}(x - \eta u)\big).$$

Based on this definition, we can rewrite the updates of Algorithm 3 as follows:

$$z_k = (1 - \alpha_{k+1})y_k + \alpha_{k+1}x_k,$$
$$x_{k+1} = x_k - \lambda_k G_{\lambda_k}(x_k, v_k),$$
$$y_{k+1} = z_k - \beta_k G_{\lambda_k}(x_k, v_k).$$

Next, we prove the following auxiliary lemma.

**Lemma 5.** *Let the sequences $\{x_k\}_k$, $\{y_k\}_k$, $\{z_k\}_k$ be generated by Algorithm 3. Then, the following inequalities hold*

$$y_k - x_k = \Gamma_k \sum_{t=1}^{k} \frac{\lambda_{t-1} - \beta_{t-1}}{\Gamma_t} G_{\lambda_{t-1}}(x_{t-1}, v_{t-1}), \tag{51}$$

$$\|y_k - x_k\|^2 \leq \Gamma_k \sum_{t=1}^{k} \frac{\lambda_{t-1} - \beta_{t-1}}{\alpha_t \Gamma_t} \|G_{\lambda_{t-1}}(x_{t-1}, v_{t-1})\|^2, \tag{52}$$

$$\|z_{k+1} - z_k\|^2 \leq 2\beta_k^2 \|G_{\lambda_k}(x_k, v_k)\|^2 + 2\alpha_{k+2}^2 \Gamma_{k+1} \sum_{t=1}^{k+1} \frac{(\lambda_{t-1} - \beta_{t-1})^2}{\alpha_t \Gamma_t} \|G_{\lambda_{t-1}}(x_{t-1}, v_{t-1})\|^2. \tag{53}$$

*Proof.* We prove the first equality. By the update rule of the momentum scheme, we obtain that

$$y_k - x_k = z_{k-1} - \beta_{k-1} G_{\lambda_{k-1}}(x_{k-1}, v_{k-1}) - (x_{k-1} - \lambda_{k-1} G_{\lambda_{k-1}}(x_{k-1}, v_{k-1}))$$
$$= (1 - \alpha_k)(y_{k-1} - x_{k-1}) + (\lambda_{k-1} - \beta_{k-1}) G_{\lambda_{k-1}}(x_{k-1}, v_{k-1}). \tag{54}$$

Dividing both sides by $\Gamma_k$ and noting that $\frac{1 - \alpha_k}{\Gamma_k} = \frac{1}{\Gamma_{k-1}}$, we further obtain that

$$\frac{y_k - x_k}{\Gamma_k} = \frac{y_{k-1} - x_{k-1}}{\Gamma_{k-1}} + \frac{\lambda_{k-1} - \beta_{k-1}}{\Gamma_k} G_{\lambda_{k-1}}(x_{k-1}, v_{k-1}). \tag{55}$$

Telescoping the above equality over $k$ yields the first desired equality.

Next, we prove the second inequality. Based on the first equality, we obtain that

$$\|y_k - x_k\|^2 = \left\| \Gamma_k \sum_{t=1}^{k} \frac{\lambda_{t-1} - \beta_{t-1}}{\Gamma_t} G_{\lambda_{t-1}}(x_{t-1}, v_{t-1}) \right\|^2$$

$$= \left\| \Gamma_k \sum_{t=1}^{k} \frac{\alpha_t}{\Gamma_t} \frac{\lambda_{t-1} - \beta_{t-1}}{\alpha_t} G_{\lambda_{t-1}}(x_{t-1}, v_{t-1}) \right\|^2$$

$$\overset{(i)}{\leq} \Gamma_k \sum_{t=1}^{k} \frac{\alpha_t}{\Gamma_t} \frac{(\lambda_{t-1} - \beta_{t-1})^2}{\alpha_t^2} \|G_{\lambda_{t-1}}(x_{t-1}, v_{t-1})\|^2$$

$$= \Gamma_k \sum_{t=1}^{k} \frac{(\lambda_{t-1} - \beta_{t-1})^2}{\Gamma_t \alpha_t} \|G_{\lambda_{t-1}}(x_{t-1}, v_{t-1})\|^2, \tag{56}$$

where (i) uses the facts that $\{\Gamma_k\}_k$ is a decreasing sequence, $\sum_{t=1}^{k} \frac{\alpha_t}{\Gamma_t} = \frac{1}{\Gamma_k}$ and Jensen's inequality.

Finally, we prove the third inequality. By the update rule of the momentum scheme, we obtain that $z_{k+1} - z_k = y_{k+1} - z_k + \alpha_{k+2}(x_{k+1} - y_{k+1})$. Then, we further obtain that

$$\|z_{k+1} - z_k\| \leq \|y_{k+1} - z_k\| + \alpha_{k+2}\|x_{k+1} - y_{k+1}\|$$

$$\leq \beta_k \|G_{\lambda_k}(x_k, v_k)\| + \alpha_{k+2}\sqrt{\|x_{k+1} - y_{k+1}\|^2}$$

$$\leq \beta_k \|G_{\lambda_k}(x_k, v_k)\| + \alpha_{k+2}\sqrt{\Gamma_{k+1} \sum_{t=1}^{k+1} \frac{(\lambda_{t-1} - \beta_{t-1})^2}{\Gamma_t \alpha_t} \|G_{\lambda_{t-1}}(x_{t-1}, v_{t-1})\|^2}.$$

The desired result follows by taking the square on both sides of the above inequality and using the fact that $(a + b)^2 \leq 2a^2 + 2b^2$. $\qquad\square$

We also need the following lemma, which was established as Lemma 1 and Proposition 1 in [11].

**Lemma 6** (Lemma 1 and Proposition 1, [11]). *Let $g$ be a proper and closed convex function. Then, for all $u, v, x \in \mathbb{R}^d$ and $\eta > 0$, the following statements hold:*

$$\langle u, G_\eta(x, u) \rangle \geq \|G_\eta(x, u)\|^2 + \frac{1}{\eta}\big(g(\text{prox}_{\eta g}(x - \eta u)) - g(x)\big),$$

$$\|G_\eta(x, u) - G_\eta(x, v)\| \leq \|u - v\|.$$

*Proof of Theorem 3*:

Consider any iteration $k$ of the algorithm. By smoothness of $f$, we obtain that

$$f(x_k) \leq f(x_{k-1}) + \langle \nabla f(x_{k-1}), x_k - x_{k-1} \rangle + \frac{L}{2}\|x_k - x_{k-1}\|^2$$

$$= f(x_{k-1}) + \langle \nabla f(x_{k-1}), -\lambda_{k-1} G_{\lambda_{k-1}}(x_{k-1}, v_{k-1}) \rangle + \frac{L\lambda_{k-1}^2}{2}\|G_{\lambda_{k-1}}(x_{k-1}, v_{k-1})\|^2$$

$$= f(x_{k-1}) - \lambda_{k-1} \langle \nabla f(x_{k-1}) - v_{k-1}, G_{\lambda_{k-1}}(x_{k-1}, v_{k-1}) \rangle - \lambda_{k-1} \langle v_{k-1}, G_{\lambda_{k-1}}(x_{k-1}, v_{k-1}) \rangle$$

$$+ \frac{L\lambda_{k-1}^2}{2}\|G_{\lambda_{k-1}}(x_{k-1}, v_{k-1})\|^2$$

$$\overset{(i)}{\leq} f(x_{k-1}) - \lambda_{k-1} \langle \nabla f(x_{k-1}) - v_{k-1}, G_{\lambda_{k-1}}(x_{k-1}, v_{k-1}) \rangle - \lambda_{k-1}\|G_{\lambda_{k-1}}(x_{k-1}, v_{k-1})\|^2$$

$$- \big(h(\text{prox}_{\lambda_{k-1}h}(x_{k-1} - \lambda_{k-1}v_{k-1})) - h(x_{k-1})\big) + \frac{L\lambda_{k-1}^2}{2}\|G_{\lambda_{k-1}}(x_{k-1}, v_{k-1})\|^2$$

$$= f(x_{k-1}) - \lambda_{k-1} \langle \nabla f(x_{k-1}) - v_{k-1}, G_{\lambda_{k-1}}(x_{k-1}, v_{k-1}) \rangle - \lambda_{k-1}\|G_{\lambda_{k-1}}(x_{k-1}, v_{k-1})\|^2$$

$$- \big(h(x_k) - h(x_{k-1})\big) + \frac{L\lambda_{k-1}^2}{2}\|G_{\lambda_{k-1}}(x_{k-1}, v_{k-1})\|^2,$$

where (i) follows from Lemma 6. Rearranging the above inequality and using Cauchy-Swartz inequality yields that

$$\Psi(x_k) \leq \Psi(x_{k-1}) - \lambda_{k-1}(1 - \frac{L\lambda_{k-1}}{2})\|G_{\lambda_{k-1}}(x_{k-1}, v_{k-1})\|^2 + \lambda_{k-1}\|\nabla f(x_{k-1}) - v_{k-1}\|\|G_{\lambda_{k-1}}(x_{k-1}, v_{k-1})\|.$$
$$(57)$$

Note that

$$\|\nabla f(x_{k-1}) - v_{k-1}\| \leq \|\nabla f(x_{k-1}) - \nabla f(z_{k-1})\| + \|\nabla f(z_{k-1}) - v_{k-1}\|$$

$$\overset{(i)}{\leq} L\|x_{k-1} - z_{k-1}\| + \|\nabla f(z_{k-1}) - v_{k-1}\|$$

$$\overset{(ii)}{\leq} L(1 - \alpha_k)\|y_{k-1} - x_{k-1}\| + \|\nabla f(z_{k-1}) - v_{k-1}\|,$$

where (i) uses the Lipschitz continuity of $\nabla f$ and (ii) follows from the update rule of the momentum scheme. Substituting the above inequality into eq. (57) yields that

$$\Psi(x_k) \leq \Psi(x_{k-1}) - \lambda_{k-1}(1 - \frac{L\lambda_{k-1}}{2})\|G_{\lambda_{k-1}}(x_{k-1}, v_{k-1})\|^2 + L\lambda_{k-1}(1 - \alpha_k)\|G_{\lambda_{k-1}}(x_{k-1}, v_{k-1})\|\|y_{k-1} - x_{k-1}\|$$

$$+ \lambda_{k-1}\|G_{\lambda_{k-1}}(x_{k-1}, v_{k-1})\|\|\nabla f(z_{k-1}) - v_{k-1}\|$$

$$\leq \Psi(x_{k-1}) - \lambda_{k-1}(1 - \frac{L\lambda_{k-1}}{2})\|G_{\lambda_{k-1}}(x_{k-1}, v_{k-1})\|^2 + \frac{L\lambda_{k-1}^2}{2}\|G_{\lambda_{k-1}}(x_{k-1}, v_{k-1})\|^2$$

$$+ \frac{L(1 - \alpha_k)^2}{2}\|y_{k-1} - x_{k-1}\|^2 + \frac{\lambda_{k-1}}{2}\|G_{\lambda_{k-1}}(x_{k-1}, v_{k-1})\|^2 + \frac{\lambda_{k-1}}{2}\|\nabla f(z_{k-1}) - v_{k-1}\|^2$$

$$= \Psi(x_{k-1}) - \lambda_{k-1}(\frac{1}{2} - L\lambda_{k-1})\|G_{\lambda_{k-1}}(x_{k-1}, v_{k-1})\|^2 + \frac{L(1 - \alpha_k)^2}{2}\|y_{k-1} - x_{k-1}\|^2$$

$$+ \frac{\lambda_{k-1}}{2}\|\nabla f(z_{k-1}) - v_{k-1}\|^2$$

$$\leq \Psi(x_{k-1}) - \lambda_{k-1}(\frac{1}{2} - L\lambda_{k-1})\|G_{\lambda_{k-1}}(x_{k-1}, v_{k-1})\|^2 + \frac{L\Gamma_{k-1}}{2}\sum_{t=1}^{k-1}\frac{\lambda_{t-1} - \beta_{t-1}}{\alpha_t \Gamma_t}\|G_{\lambda_{t-1}}(x_{t-1}, v_{t-1})\|^2$$

$$+ \frac{\lambda_{k-1}}{2}\|\nabla f(z_{k-1}) - v_{k-1}\|^2,$$

where the last inequality uses item 2 of Lemma 5 and the fact that $0 < \alpha_k < 1$. Telescoping the above inequality over $k$ from 1 to $K$ yields that

$$\Psi(x_K) \leq \Psi(x_0) - \sum_{k=0}^{K-1} \lambda_k \left(\frac{1}{2} - L\lambda_k\right) \|G_{\lambda_k}(x_k, v_k)\|^2 + \sum_{k=0}^{K-1} \frac{L\Gamma_k}{2} \sum_{t=0}^{k-1} \frac{(\lambda_t - \beta_t)^2}{\Gamma_{t+1}\alpha_{t+1}} \|G_{\lambda_t}(x_t, v_t)\|^2$$

$$+ \sum_{k=0}^{K-1} \frac{\lambda_k}{2} \|\nabla f(z_k) - v_k\|^2$$

$$= \Psi(x_0) - \sum_{k=0}^{K-1} \lambda_k \left(\frac{1}{2} - L\lambda_k\right) \|G_{\lambda_k}(x_k, v_k)\|^2 + \frac{L}{2} \sum_{k=0}^{K-1} \frac{(\lambda_k - \beta_k)^2}{\Gamma_{k+1}\alpha_{k+1}} \|G_{\lambda_k}(x_k, v_k)\|^2 \left(\sum_{t=k}^{K-1} \Gamma_t\right)$$

$$+ \sum_{k=0}^{K-1} \frac{\lambda_k}{2} \|\nabla f(z_k) - v_k\|^2, \tag{58}$$

where we have exchanged the order of summation in the second equality. Furthermore, note that $\sum_{t=k}^{K-1} \Gamma_t = 2\sum_{t=k}^{K-1} \frac{1}{\lceil t/q \rceil} - \frac{1}{\lceil t/q \rceil + 1} \leq \frac{2}{\lceil k/q \rceil}$. Then, substituting this bound into the above inequality and taking expectation on both sides yield that

$$\mathbb{E}[\Psi(x_K)] \leq \Psi(x_0) - \sum_{k=0}^{K-1} \lambda_k \left(\frac{1}{2} - L\lambda_k\right) \mathbb{E}\|G_{\lambda_k}(x_k, v_k)\|^2 + \frac{L}{2} \sum_{k=0}^{K-1} \frac{2(\lambda_k - \beta_k)^2}{\lceil k/q \rceil \Gamma_{k+1}\alpha_{k+1}} \mathbb{E}\|G_{\lambda_k}(x_k, v_k)\|^2$$

$$+ \sum_{k=0}^{K-1} \frac{\lambda_k}{2} \mathbb{E}\|\nabla f(z_k) - v_k\|^2. \tag{59}$$

Next, we bound the term $\mathbb{E}\|\nabla f(z_k) - v_k\|^2$ in the above inequality. By Lemma 4 we obtain that

$$\mathbb{E}\|\nabla f(z_k) - v_k\|^2 \leq \sum_{i=(\tau(k)-1)q}^{k-1} \frac{L^2}{|\xi_i|} \mathbb{E}\|z_{i+1} - z_i\|^2$$

$$\leq \sum_{i=(\tau(k)-1)q}^{k-1} \frac{L^2}{|\xi_i|} \left[2\beta_i^2 \|G_{\lambda_i}(x_i, v_i)\|^2 + 2\alpha_{i+2}^2 \Gamma_{i+1} \sum_{t=0}^{i} \frac{(\lambda_t - \beta_t)^2}{\alpha_{t+1}\Gamma_{t+1}} \|G_{\lambda_t}(x_t, v_t)\|^2\right], \tag{60}$$

where the last inequality uses item 3 of Lemma 5. Substituting eq. (60) into eq. (59) and simplifying yield that

$$\mathbb{E}[\Psi(x_K)] \leq \Psi(x_0) - \sum_{k=0}^{K-1} \left[\lambda_k \left(\frac{1}{2} - L\lambda_k\right) - \frac{L(\lambda_k - \beta_k)^2}{\lceil k/q \rceil \Gamma_{k+1}\alpha_{k+1}}\right] \mathbb{E}\|G_{\lambda_k}(x_k, v_k)\|^2$$

$$+ \underbrace{\sum_{k=0}^{K-1} \frac{\lambda_k}{2} \mathbb{E}\left[\sum_{i=(\tau(k)-1)q}^{k-1} \frac{L^2}{|\xi_i|} \left[2\beta_i^2 \|G_{\lambda_i}(x_i, v_i)\|^2 + 2\alpha_{i+2}^2 \Gamma_{i+1} \sum_{t=0}^{i} \frac{(\lambda_t - \beta_t)^2}{\alpha_{t+1}\Gamma_{t+1}} \|G_{\lambda_t}(x_t, v_t)\|^2\right]\right]}_{T}. \tag{61}$$

Before we proceed the proof, we first specify the choices of all the parameters. Specifically, we choose a constant mini-batch size $|\xi_k| \equiv |\xi|$, a constant $q = |\xi|$, a constant $\beta_k \equiv \beta > 0$, $\lambda_k \in [\beta, (1 + \alpha_{k+1})\beta]$. Based on these parameter settings, the term $T$ in the above inequality can be

bounded as follows.

$$
\begin{aligned}
T &\overset{(i)}{\leq} \sum_{k=0}^{K-1} \frac{\lambda_k}{2} \mathbb{E}\Bigg[ \sum_{i=(\tau(k)-1)q}^{\tau(k)q-1} \frac{L^2}{|\xi_i|} \Big[ 2\beta_i^2 \|G_{\lambda_i}(x_i, v_i)\|^2 + 2\alpha_{i+2}^2 \Gamma_{i+1} \sum_{t=0}^{k-1} \frac{(\lambda_t - \beta_t)^2}{\alpha_{t+1}\Gamma_{t+1}} \|G_{\lambda_t}(x_t, v_t)\|^2 \Big] \Bigg] \\
&\overset{(ii)}{\leq} \sum_{k=0}^{K-1} \frac{\lambda_k L^2 q \beta^2}{|\xi|} \mathbb{E}\|G_{\lambda_k}(x_k, v_k)\|^2 + \sum_{k=0}^{K-1} \frac{2\lambda_k L^2}{|\xi|\tau(k)^3} \sum_{t=0}^{k-1} \frac{(\lambda_t - \beta_t)^2}{\alpha_{t+1}\Gamma_{t+1}} \mathbb{E}\|G_{\lambda_t}(x_t, v_t)\|^2 \\
&\overset{(iii)}{\leq} \sum_{k=0}^{K-1} \lambda_k L^2 \beta^2 \mathbb{E}\|G_{\lambda_k}(x_k, v_k)\|^2 + \frac{2L^2\beta^2}{|\xi|} \sum_{k=0}^{K-1} \frac{\alpha_{k+1}}{\Gamma_{k+1}} \mathbb{E}\|G_{\lambda_k}(x_k, v_k)\|^2 \Big( \sum_{t=k}^{K-1} \frac{\lambda_k}{\tau(t)^3} \Big) \\
&\overset{(iv)}{\leq} \sum_{k=0}^{K-1} \lambda_k L^2 \beta^2 \mathbb{E}\|G_{\lambda_k}(x_k, v_k)\|^2 + \frac{4L^2\beta^3}{|\xi|} \sum_{k=0}^{K-1} (\lceil k/q \rceil + 1) \mathbb{E}\|G_{\lambda_k}(x_k, v_k)\|^2 \Big( \sum_{t=(\tau(k)-1)q}^{\tau(K)q} \frac{1}{\tau(t)^3} \Big) \\
&= \sum_{k=0}^{K-1} \lambda_k L^2 \beta^2 \mathbb{E}\|G_{\lambda_k}(x_k, v_k)\|^2 + \frac{4L^2\beta^3}{|\xi|} \sum_{k=0}^{K-1} (\lceil k/q \rceil + 1) \mathbb{E}\|G_{\lambda_k}(x_k, v_k)\|^2 \Big( \sum_{t=\tau(k)-1}^{\tau(K)} \frac{q}{(t+1)^3} \Big) \\
&\leq \sum_{k=0}^{K-1} \lambda_k L^2 \beta^2 \mathbb{E}\|G_{\lambda_k}(x_k, v_k)\|^2 + 2L^2\beta^3 \sum_{k=0}^{K-1} (\lceil k/q \rceil + 1) \mathbb{E}\|G_{\lambda_k}(x_k, v_k)\|^2 \frac{1}{\tau(k)^2} \\
&\overset{(v)}{\leq} \sum_{k=0}^{K-1} \lambda_k L^2 \beta^2 \mathbb{E}\|G_{\lambda_k}(x_k, v_k)\|^2 + 2L^2\beta^3 \sum_{k=0}^{K-1} \mathbb{E}\|G_{\lambda_k}(x_k, v_k)\|^2 \frac{\lceil k/q \rceil + 1}{\tau(k)^2} \\
&\leq \sum_{k=0}^{K-1} \lambda_k L^2 \beta^2 \mathbb{E}\|G_{\lambda_k}(x_k, v_k)\|^2 + 2L^2\beta^3 \sum_{k=0}^{K-1} \mathbb{E}\|G_{\lambda_k}(x_k, v_k)\|^2,
\end{aligned}
\tag{62}
$$

where (i) follows from the facts that $i \leq k-1$ and $k-1 \leq \tau(k)q - 1$, (ii) uses the fact that $\sum_{i=(\tau(k)-1)q}^{\tau(k)q-1} \alpha_{i+2}^2 \Gamma_{i+1} \leq \frac{2}{\tau(k)^3}$, (iii) uses the parameter settings $q = |\xi|$ and $\lambda_t - \beta_t \leq \alpha_t\beta$, (iv) uses the facts that $\lambda_k \leq 2\beta$ and $(\tau(k) - 1)q \leq k \leq \tau(k)q$ and (v) uses the fact that $k \leq \tau(k)q - 1$. Substituting the above inequality into eq. (61) and simplifying, we obtain that

$$
\mathbb{E}[\Psi(x_K)] \leq \Psi(x_0) - \sum_{k=0}^{K-1} \Big[ \lambda_k \Big( \frac{1}{2} - L\lambda_k - L^2\beta^2 \Big) - \frac{L(\lambda_k - \beta_k)^2}{\lceil k/q \rceil \Gamma_{k+1}\alpha_{k+1}} - 2L^2\beta^3 \Big] \mathbb{E}\|G_{\lambda_k}(x_k, v_k)\|^2
\tag{63}
$$

$$
\leq \Psi(x_0) - \sum_{k=0}^{K-1} \Big[ \beta\Big( \frac{1}{2} - 2L\beta - L^2\beta^2 \Big) - L\beta^2 - 2L^2\beta^3 \Big] \mathbb{E}\|G_{\lambda_k}(x_k, v_k)\|^2.
\tag{64}
$$

Choosing $\beta \leq \frac{1}{8L}$, the above inequality further implies that

$$
\mathbb{E}[\Psi(x_K)] \leq \Psi(x_0) - \sum_{k=0}^{K-1} \frac{\beta}{16} \mathbb{E}\|G_{\lambda_k}(x_k, v_k)\|^2.
\tag{65}
$$

Then, it follows that $\frac{1}{K} \sum_{k=0}^{K-1} \mathbb{E}\|G_{\lambda_k}(x_k, v_k)\|^2 \leq 16(\Psi(x_0) - \Psi^*)/(K\beta)$. Next, we bound the term $\mathbb{E}\|G_{\lambda_\zeta}(z_\zeta, \nabla f(z_\zeta))\|^2$, where $\zeta$ is selected uniformly at random from $\{0, \dots, K-1\}$. Observe

that

$$
\begin{aligned}
\mathbb{E}\|G_{\lambda_\zeta}(z_\zeta, \nabla f(z_\zeta))\|^2 &= \mathbb{E}\|G_{\lambda_\zeta}(z_\zeta, \nabla f(z_\zeta)) - G_{\lambda_\zeta}(z_\zeta, v_\zeta) + G_{\lambda_\zeta}(z_\zeta, v_\zeta)\|^2 \\
&\leq 2\mathbb{E}\|G_{\lambda_\zeta}(z_\zeta, \nabla f(z_\zeta)) - G_{\lambda_\zeta}(z_\zeta, v_\zeta)\|^2 + 2\mathbb{E}\|G_{\lambda_\zeta}(z_\zeta, v_\zeta)\|^2 \\
&\overset{(i)}{\leq} 2\mathbb{E}\|\nabla f(z_\zeta) - v_\zeta\|^2 + 2\mathbb{E}\|G_{\lambda_\zeta}(z_\zeta, v_\zeta) - G_{\lambda_\zeta}(x_\zeta, v_\zeta) + G_{\lambda_\zeta}(x_\zeta, v_\zeta)\|^2 \\
&\leq 2\mathbb{E}\|\nabla f(z_\zeta) - v_\zeta\|^2 + 4\mathbb{E}\|G_{\lambda_\zeta}(z_\zeta, v_\zeta) - G_{\lambda_\zeta}(x_\zeta, v_\zeta)\|^2 + 4\mathbb{E}\|G_{\lambda_\zeta}(x_\zeta, v_\zeta)\|^2 \\
&\leq 2\mathbb{E}\|\nabla f(z_\zeta) - v_\zeta\|^2 + 4\mathbb{E}\|G_{\lambda_\zeta}(x_\zeta, v_\zeta)\|^2 \\
&\quad + \frac{4}{\lambda_\zeta^2}\mathbb{E}\|z_\zeta - x_\zeta + \mathrm{prox}_{\lambda_\zeta h}(x_\zeta - \lambda_\zeta v_\zeta) - \mathrm{prox}_{\lambda_\zeta h}(z_\zeta - \lambda_\zeta v_\zeta)\|^2 \\
&\leq 2\mathbb{E}\|\nabla f(z_\zeta) - v_\zeta\|^2 + 4\mathbb{E}\|G_{\lambda_\zeta}(x_\zeta, v_\zeta)\|^2 \\
&\quad + \frac{8}{\lambda_\zeta^2}\mathbb{E}\|z_\zeta - x_\zeta\|^2 + \frac{8}{\lambda_\zeta^2}\mathbb{E}\|\mathrm{prox}_{\lambda_\zeta h}(x_\zeta - \lambda_\zeta v_\zeta) - \mathrm{prox}_{\lambda_\zeta h}(z_\zeta - \lambda_\zeta v_\zeta)\|^2 \\
&\overset{(ii)}{\leq} 2\mathbb{E}\|\nabla f(z_\zeta) - v_\zeta\|^2 + 4\mathbb{E}\|G_{\lambda_\zeta}(x_\zeta, v_\zeta)\|^2 + \frac{8}{\lambda_\zeta^2}\mathbb{E}\|z_\zeta - x_\zeta\|^2 + \frac{8}{\lambda_\zeta^2}\mathbb{E}\|z_\zeta - x_\zeta\|^2 \\
&\overset{(iii)}{\leq} 2\mathbb{E}\|\nabla f(z_\zeta) - v_\zeta\|^2 + 4\mathbb{E}\|G_{\lambda_\zeta}(x_\zeta, v_\zeta)\|^2 + \frac{16}{\lambda_\zeta^2}\mathbb{E}\|y_\zeta - x_\zeta\|^2 \qquad (66)
\end{aligned}
$$

where (i) uses the non-expansiveness property of the operator $G$ in Lemma 6, (ii) follows from the non-expansiveness of the proximal operator, and (iii) uses the update rule and the fact that $0 < \alpha_k < 1$.

Next, we bound the three terms on the right hand side of the above inequality separately. First, note that

$$
\mathbb{E}\|G_{\lambda_\zeta}(x_\zeta, v_\zeta)\|^2 = \frac{1}{K}\sum_{k=0}^{K-1}\mathbb{E}\|G_{\lambda_k}(x_k, v_k)\|^2 \leq \frac{16(\Psi(x_0) - \Psi^*)}{K\beta}.
$$

Second, note that eq. (60) implies that

$$
\begin{aligned}
&\mathbb{E}\|\nabla f(z_\zeta) - v_\zeta\|^2 \\
&\leq \mathbb{E}\sum_{i=(\tau(\zeta)-1)q}^{\zeta-1} \frac{L^2}{|\xi_i|}\left[2\beta_i^2\|G_{\lambda_i}(x_i, v_i)\|^2 + 2\alpha_{i+2}^2\Gamma_{i+1}\sum_{t=0}^{i}\frac{(\lambda_t - \beta_t)^2}{\alpha_{t+1}\Gamma_{t+1}}\|G_{\lambda_t}(x_t, v_t)\|^2\right] \\
&\leq \frac{2L^2\beta^2}{|\xi|}\mathbb{E}\left(\sum_{i=(\tau(\zeta)-1)q}^{\tau(\zeta)q-1}\|G_{\lambda_i}(x_i, v_i)\|^2\right) + \frac{L^2}{|\xi|}\mathbb{E}\left(\sum_{i=(\tau(\zeta)-1)q}^{\zeta-1} 2\alpha_{i+2}^2\Gamma_{i+1}\sum_{t=0}^{i}\frac{(\lambda_t - \beta_t)^2}{\alpha_{t+1}\Gamma_{t+1}}\|G_{\lambda_t}(x_t, v_t)\|^2\right) \\
&\leq \frac{2L^2\beta^2}{|\xi|K}\sum_{\zeta=0}^{K-1}\left(\sum_{i=(\tau(\zeta)-1)q}^{\tau(\zeta)q-1}\mathbb{E}\|G_{\lambda_i}(x_i, v_i)\|^2\right) + \frac{L^2\beta^2}{|\xi|K}\sum_{\zeta=0}^{K-1}\left(\sum_{i=(\tau(\zeta)-1)q}^{\tau(\zeta)q-1} 2\alpha_{i+2}^2\Gamma_{i+1}\sum_{t=0}^{\zeta-1}(t+1)\mathbb{E}\|G_{\lambda_t}(x_t, v_t)\|^2\right) \\
&\leq \frac{2L^2\beta^2 q}{|\xi|}\frac{1}{K}\sum_{\zeta=0}^{K-1}\mathbb{E}\|G_{\lambda_\zeta}(x_\zeta, v_\zeta)\|^2 + \frac{L^2\beta^2}{|\xi|}\frac{1}{K}\sum_{\zeta=0}^{K-1}\left(\frac{4}{\tau(\zeta)^3}\sum_{t=0}^{\zeta-1}(\lceil t/q \rceil + 1)\mathbb{E}\|G_{\lambda_t}(x_t, v_t)\|^2\right) \\
&\leq 2L^2\beta^2\left(\frac{1}{K}\sum_{\zeta=0}^{K-1}\mathbb{E}\|G_{\lambda_\zeta}(x_\zeta, v_\zeta)\|^2\right) + \frac{L^2\beta^2}{|\xi|}\frac{1}{K}\sum_{\zeta=0}^{K-1}(\lceil \zeta/q \rceil + 1)\mathbb{E}\|G_{\lambda_\zeta}(x_\zeta, v_\zeta)\|^2\sum_{t=\zeta}^{K-1}\frac{4}{\tau(t)^3} \\
&\leq 2L^2\beta^2\left(\frac{1}{K}\sum_{\zeta=0}^{K-1}\mathbb{E}\|G_{\lambda_\zeta}(x_\zeta, v_\zeta)\|^2\right) + L^2\beta^2\frac{1}{K}\sum_{\zeta=0}^{K-1}\mathbb{E}\|G_{\lambda_\zeta}(x_\zeta, v_\zeta)\|^2\frac{2(\lceil \zeta/q \rceil + 1)}{\tau(\zeta)]^2} \\
&\leq 3L^2\beta\frac{16(\Psi(x_0) - \Psi^*)}{K},
\end{aligned}
$$

where we have used the fact that $\zeta$ is sampled uniformly from $0, ..., K-1$ at random.

Third, note that by item 2 of Lemma 5, we know that

$$
\begin{aligned}
\mathbb{E}\|y_\zeta - x_\zeta\|^2 &\le \mathbb{E}\left(\Gamma_\zeta \sum_{t=0}^{\zeta-1} \frac{\lambda_t - \beta_t}{\alpha_{t+1}\Gamma_{t+1}} \|G_{\lambda_t}(x_t, v_t)\|^2\right) \\
&\le \frac{1}{K} \sum_{\zeta=0}^{K-1} \Gamma_\zeta \sum_{t=0}^{\zeta-1} \frac{\lambda_t - \beta_t}{\alpha_{t+1}\Gamma_{t+1}} \mathbb{E}\|G_{\lambda_t}(x_t, v_t)\|^2 \\
&\le \frac{\beta^2}{K} \sum_{\zeta=0}^{K-1} \Gamma_\zeta \sum_{t=0}^{\zeta-1} (\lceil t/q \rceil + 1) \mathbb{E}\|G_{\lambda_t}(x_t, v_t)\|^2 \\
&= \frac{\beta^2}{K} \sum_{\zeta=0}^{K-1} (\lceil \zeta/q \rceil + 1) \mathbb{E}\|G_{\lambda_\zeta}(x_\zeta, v_\zeta)\|^2 \left(\sum_{t=\zeta+1}^{K-1} \Gamma_t\right) \\
&\le \frac{\beta^2}{K} \sum_{\zeta=0}^{K-1} \mathbb{E}\|G_{\lambda_\zeta}(x_\zeta, v_\zeta)\|^2.
\end{aligned}
\tag{67}
$$

Combining the above three inequalities and note that $L\beta = \Theta(1)$ and $\frac{\beta}{\lambda_k} \le 1$, we finally obtain that

$$
\mathbb{E}\|G_{\lambda_\zeta}(z_\zeta, \nabla f(z_\zeta))\|^2 \le \mathcal{O}\left(\frac{L(\Psi(x_0) - \Psi^*)}{K}\right).
\tag{68}
$$

This further implies that

$$
\mathbb{E}\|G_{\lambda_\zeta}(z_\zeta, \nabla f(z_\zeta))\| \le \sqrt{\mathbb{E}\|G_{\lambda_\zeta}(z_\zeta, \nabla f(z_\zeta))\|^2} \le \mathcal{O}\left(\sqrt{\frac{L(\Psi(x_0) - \Psi^*)}{K}}\right).
$$

Setting the right hand side of the above inequality to be bounded by $\epsilon$, we obtain that $K \ge \mathcal{O}\left(\frac{L(\Psi(x_0) - \Psi^*)}{\epsilon^2}\right)$. Then, the total number of stochastic gradient calls is bounded by $(K + q)\frac{n}{q} + K|\xi| \le \mathcal{O}(n + \sqrt{n}\epsilon^{-2})$.

## J.2   Proof of Theorem 8

The proof follows exactly from that of Theorem 3 (the same treatment of the momentum schemes applies).