[Reviews · NeurIPS 2019]

Reviewer 1



The paper contains technically novel contributions and comprehensive theoretical analysis that cover many important scenarios. It is also well-written. Thus, I recommend to accept the paper for publication in NeurIPS. Below are my related questions: 1. SpiderBoost uses sampling with replacement. As it is known that sampling without replacement induces lower sampling variance, can similar analysis be applied to handle such a sampling scheme and obtain better complexity result? 2. The final output is randomly selected among all past iterations. Does this require to store all the generated variables?

Reviewer 2



Apologies I reviewed all three papers in one document and then cut and past the comments in. Here are my comments specific to `SpiderBoost and Momentum: Faster Variance Reduction Algorithms' *Line 273. Which recent studies? *Why did the numerics in `SpiderBoost and Momentum: Faster Stochastic Variance Reduction Algorithms’ not include MNIST? From Figure 3 of `Faster Stochastic Algorithms via History-Gradient Aided Batch Size Adaptation’ I can see that spiderboost is not doing well relative to mini batch SGD. Don’t be afraid to include results where your algorithm does not look good. It is interesting to discuss why (you can for example point out that the cause is perhaps large batch sizes). *Table 1. Why \eps^{-2} + \eps^{-3}? Just write \eps^{-3}. *It appears test error was not included in the numerics. *I’d suggest including a standard widely used algorithm in the comparisons, e.g., ADAM. * Big-O notation is not used consistently throughout the three papers. For example, in Theorem 1 of 1410 bounds K including Lipschitz constants and function value but then the SFO complexity ignores Lipschitz constants and function value which should be included in the expression. As a general rule of thumb, except in an introduction big-O notation should include all dependencies on Lipschitz constants, etc. * Line 111. In the context of this paper (since the gradient is not rescaled) the step size of spider is actually O( \eps / (L \| \grad f \|) ). To see this, compare line 11 and 14 of algorithm 1 of the spider paper with algorithm 1 in this paper. The step size selection of the original spider is poor when \eps / \| \grad f \| is much less than one. * Line 46-50. I don't understand this reasoning. The proximal mapping is non-expansive which appears to contradict your logic. Reviewer comment: `For example, creating one long, high quality paper’. Authors response to reviewer: [authors were worried the paper would be too long]. This is just one recommendation, in this case one would probably best to send it to a journal. Also, I am not suggesting copying and pasting the papers together but merging them into one coherent framework and removing a large number of results that are not core to your thesis. You could also merge 1410 and 2460 into one paper. These are just suggestions it is up to the authors to figure out the best strategy. Authors: [1495 is very different] Yes I agree with the statement that 1495 is more different than 1410 and 2460 are from each other, but they all still share a lot of ideas and similar theoretical results. Reviewer: `All three papers develop a practical **variant** of SPIDER algorithm which achieves the same bounds as SPIDER.’ Authors response to above comment: `First, not all three papers focus on SPIDER algorithms. The other two papers study other aspects of stochastic algorithms (paper 2460 on batch size adaptation, and paper 1495 on reinforcement learning) and both analyze three representative stochastic algorithms SGD, SVRG and SPIDER as examples.’ The keyword in my response is variant which I have starred. SVRG is also extremely strongly similar to SPIDER. The main distinguishing feature of the original SPIDER paper was the better theoretical runtime bound. Paper 1410 specific response to author: Reviewer `PL results are straightforward corollaries of main results. I would suggest removing them.’ Authors: `This is not true! PL results are linear rate obtained under PL condition, whereas the main results are sublinear rate obtained for nonconvex optimization. Clearly, PL results cannot be a special case of the main results.’ You misinterpreted what I meant. This however is my fault as I should have been more clear. Given an algorithm with a sublinear convergence rate of the form (f(x_k) - f(x_*)) / \eps^{2} for the gradient and using the PL inequality there is a very standard (and short) argument to obtain linear convergence. In my opinion there are way to many papers on this (the proof is extremely similar to the linear convergence proof for gradient descent). For this reason, I don’t think the result is very original and detracts from the main results.

Reviewer 3



Pros. + This paper is well written and easy to follow + This paper builds on recently proposed algorithms SARAH and SPYDER for non-convex problems. Its main contribution is a novel analysis, which allows stepsize to be constant O(1/L) rather than to be dependent on the desired accuracy \epsilon. + authors provide different extensions of their method – extension to proximal settings, momentum and methods for online optimization. + all the rates outperform or match the current state-of-the-art Cons. - constant step size 1/2L is enabled due to big minibatch \sqrt{n}, which could be a problem for the practical usage of this new method. - Experimental results are not appealing since the parameters are chosen neither based on theory nor any tuning. Moreover, there is no comparison to SAGA algorithm, which would be also a natural choice. Minor: Alg 3. - Prox (x_k - \lambda_k v_k) is used twice, while the second usage might be replaced by x_{k+1} *** I have read the authors response and other reviews and decided to keep my score. I strongly reccomend authors to include minibatch version and to remove stepsize of 1/2L from the definition of the algorithm to avoid confusion and make their work more transparent.

[Author Response · NeurIPS 2019]

Since reviewer 3 raised a most critical issue, we respond to this reviewer first.

**Reviewer 3: Q:** *The three papers are extremely similar.*

**A:** We are so shocked about the reviewer's judgement. The contributions of three papers are fully orthogonal, and
completely under three different research lines/topics: **paper 1410 (this paper)** on proximal and momentum algorithms,
**paper 2460** on batch size adaptation, and **paper 1495** on reinforcement learning (RL) algorithms. In the past, there
have been extensive studies under each research line, and many top conference papers contribute only to one line.

**Q:** *All three papers focus on developing a practical variant of SPIDER algorithm which achieves same theoretical*
*bounds as SPIDER.*

**A:** First, not all three papers focus on SPIDER algorithms. Only paper 1410 focuses on proximal and acceleration of
SPIDER. The other two papers study other aspects of stochastic algorithms (paper 2460 on batch size adaptation, and
paper 1495 on reinforcement learning) and both analyze three representative stochastic algorithms SGD, SVRG and
SPIDER as examples. Inclusion of SPIDER in these two papers is mainly because it is the state-of-the-art stochastic
algorithm. Even for SPIDER-type algorithms, analyzing them under three directions still requires significantly different
treatments. They do not achieve the same theoretical bound either.

**Specifically for this paper 1410, we emphasize its difference from the other two papers below.**

• This paper makes two major contributions: develop and prove that **proximal** SPIDER algorithm has better
complexity order than existing art; and develop **momentum** SPIDER algorithm and prove its performance guarantee.
Neither of the other two papers even touch the topics of proximal and momentum algorithms.
• The performance guarantee for both algorithms in this paper are nontrivial, and require considerable new technical
developments specifically for proximal and momentum algorithms, which were absolutely not in the other papers'
proofs. In fact, the proof of vanilla SPIDER does not extend to Proximal SPIDER.

**Q:** *To increase the score: (1) At the very least each of your papers should devote a paragraph to explaining the distinct*
*contributions of each paper and ideally how they form a cohesive body. (2) Personally, I'd suggest reducing the number*
*of papers. For example, creating one long, high quality paper.*

**A:** Regarding (1), both papers 2460 and 1495 have already discussed their differences from paper 1410. Paper 1410 did
not discuss the other two papers because they have not been released publicly yet.

Regarding (2), the three papers are developed under different research lines and should naturally be written separately.
**Specifically for this paper 1410, we have already made the best efforts into including as many relevant results as**
**possible to supplementary materials, which we believe has made a comprehensive body of work by itself.** Even
if we try to combine them, it is unrealistic for NeurIPS. Clearly, the reviewer also realizes combining all three papers
would be LONG, and 8-page limit of NeurIPS submission will not allow all major results to be presented within the
main text.

**Q:** *Detailed comments*

**A:** It is disappointing that the reviewer's detailed comments posted here are not for this paper 1410, but for paper 2460.

**Reviewer 1: Q1:** *Can similar analysis be applied to handle sampling without replacement and obtain better complexity?*

**A:** Yes, the analysis here can be applied to sampling without replacement, but need to incorporate different concentration
bounds due to the sampling without replacement. We expect the resulting bounds to be tighter.

**Q2:** *The final output is randomly selected among all past iterations. Does this require to store all generated variables?*

**A:** The implementation of the algorithm does not need to store all variables. Our theory suggests that we can randomly
pick $i \in 1, \cdots K$ first, and then stop at the $i$-th iteration. In practice, we can store only the variable with minimum batch
gradient norm. Alternatively, outputting the last iteration also performs well in practice.

**Reviewer 4: Q1:** *Step size 1/2L is enabled due to big minibatch $\sqrt{n}$, which could be a problem for its practical usage.*

**A:** We agree. This is also the reason why we study online version (see appendix), which requires the minibatch $\mathcal{O}(\epsilon^{-1})$
and total sample complexity of $\mathcal{O}(\epsilon^{-3})$ to achieve the same accuracy. This helps for large $n$ regime.

**Q2:** *Experimental results are not appealing since the parameters are chosen neither based on theory nor any tuning.*
*Moreover, there is no comparison to SAGA algorithm, which would be also a natural choice.*

**A:** Thanks for the suggestion! We will add more experiments as suggested.

**Q3, Q4, Q5:** *presentation suggestions*

**A:** Thanks for the suggestions! We will revise the paper accordingly.

[Meta-Review · NeurIPS 2019]

There was significant discussion among the reviewers regarding the 3 papers sharing authors that explored variants of SPIDER (among other things). In this particular case, the reviewers think this is a real step forward towards developing a practical variant. Nevertheless, the reviewers have various suggestions that should be taken into account, that would improve the quality of the paper.